# REAL: Regression-Aware Reinforcement Learning for LLM-as-a-Judge

Yasi Zhang [1] [*]    Tianyu Chen [2] [*]    Mingyuan Zhou [2]    Oscar Leong [1]    Ying Nian Wu [1]    Michal Lukasik [3] [†]

## Abstract

Large language models (LLMs) are increasingly deployed as automated evaluators that assign numeric scores to model outputs, a paradigm known as LLM-as-a-Judge. However, standard Reinforcement Learning (RL) methods typically rely on binary rewards (e.g., 0-1 accuracy), thereby ignoring the ordinal structure inherent in regression tasks; for instance, they fail to recognize that predicting 4 is significantly better than predicting 1 when the ground truth is 5. Conversely, existing regression-aware approaches are often confined to Supervised Fine-Tuning (SFT), limiting their ability to explore optimal reasoning paths. To bridge this gap, we propose **REAL** (**RE**gression-**A**ware Reinforcement **L**earning), a principled RL framework designed to optimize regression rewards, and also proven to be optimal for correlation metrics. A key technical challenge is that the regression objective is explicitly policy-dependent, thus invalidating standard policy gradient methods. To address this, we employ the generalized policy gradient estimator, which naturally decomposes optimization into two complementary components: (1) exploration over Chain-of-Thought (CoT) trajectory, and (2) regression-aware prediction refinement of the final score. Extensive experiments across model scales (8B to 32B) demonstrate that REAL consistently outperforms both regression-aware SFT baselines and standard RL methods, exhibiting significantly better generalization on out-of-domain benchmarks. These findings highlight the critical value of integrating regression objectives into RL exploration for accurate LLM evaluation. Code is available at https://github.com/YasminZhang/REAL.

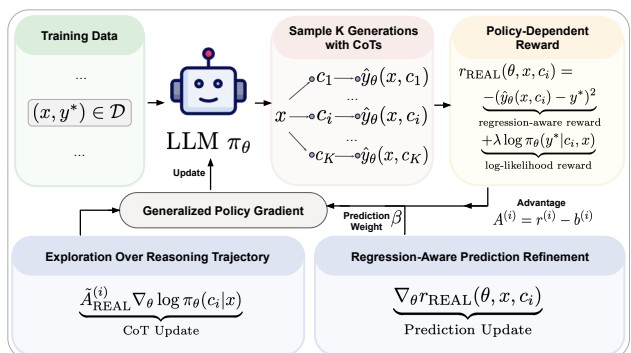

*Figure 1.* **Overview of the REAL framework.** REAL addresses the limitations of standard RL in LLM-as-a-Judge tasks by optimizing a policy-dependent regression reward. The framework employs a generalized policy gradient that leads to a gradient update that decomposes into two terms: (1) **Exploration Over Reasoning Trajectory**; and (2) **Regression-Aware Prediction Refinement**. This enables principled optimization of ordinal structures that standard RL with binary rewards typically ignores. The full algorithm is in Alg. 2.

## 1. Introduction

Large Language Models (LLMs) (Brown et al., 2020; Achiam et al., 2023) have evolved beyond mere content generators to become sophisticated evaluators, a paradigm known as LLM-as-a-Judge (Gu et al., 2024; Chiang & Lee, 2023). This role is now central to diverse applications, from assessing text quality (Ouyang et al., 2022) and instruction following (Zhu et al., 2025) to safety alignment (Zhu et al., 2024) and preference modeling (Kim et al., 2023). In these contexts, the model must produce a numeric score that accurately reflects quality, correctness, or preference intensity.

Despite the regression nature of these tasks, standard practices—exemplified by Prometheus 1 and 2 (Kim et al., 2023; 2024)—rely on traditional SFT via cross-entropy loss. By treating numeric scores as categorical tokens, these methods ignore the inherent ordinal structure of the data. Recent advancements, such as Regression-Aware Fine-Tuning (RAFT) (Lukasik et al., 2025; 2024), have begun to bridge this gap by optimizing models using regression losses over expected numerical predictions. Extensions like TRACT (Chiang et al., 2025) further incorporate Chain-of-Thought (CoT) supervision to improve reasoning faithfulness. How-

---
[*]Equal contribution  [1]University of California, Los Angeles [2]The University of Texas at Austin [3]Google Research [†]Now at Google DeepMind. Correspondence to: Yasi Zhang <yasminzhang@ucla.edu>, Tianyu Chen <tianyuchen@utexas.edu>.

*Proceedings of the $43^{rd}$ International Conference on Machine Learning*, Seoul, South Korea. PMLR 306, 2026. Copyright 2026 by the author(s).

ever, these approaches remain confined to the SFT regime: they depend on static ground-truth trajectories and lack a principled mechanism for exploring model-generated reasoning pathways.

Transitioning from regression-aware SFT to Reinforcement Learning (RL) is, therefore, a natural progression to enable exploration over CoT pathways. This shift allows the model to search for optimal reasoning trajectories guided by downstream numerical objectives. However, standard RL post-training frameworks (Schulman et al., 2017; Guo et al., 2025; Ahmadian et al., 2024) typically rely on rule-based verifiers to produce binary rewards (Guo et al., 2025) (e.g., 0–1 accuracy), creating a mismatch between the LLM-as-a-Judge task's regression nature and categorical RL optimization. For instance, standard RL fails to recognize that predicting a 4 is significantly better than predicting a 1 when the ground truth is 5. Empirically, we also find that standard RL leads to suboptimal learning behavior, specifically a collapse in correlation metrics (see Fig. 2 and Tab. 2).

To address this challenge, we introduce Regression-Aware Reinforcement Learning (REAL), a principled framework designed specifically for LLM-as-a-Judge and pointwise evaluation tasks, as presented in Fig. 1. However, integrating regression-aware objectives into RL introduces a significant theoretical challenge: the resulting reward functions depend explicitly on the policy parameters themselves. To resolve the resulting policy-dependency reward issue, we employ the generalized policy gradient (Schulman et al., 2015) that explicitly accounts for parameter-dependent rewards. This derivation decomposes the learning process into two complementary components: (1) A CoT policy-gradient term that encourages exploration over reasoning trajectories based on downstream regression-aware rewards; and (2) A prediction refinement term that provides regression-aware supervision to the final numerical prediction via standard backpropagation. This decomposition clarifies how reasoning exploration and numerical accuracy can be jointly optimized within a unified framework.

Theoretically, Sec. 3 proves that minimizing the regression loss leads to optimal optimization in terms of correlation metrics. Empirically, REAL consistently outperforms prior regression-aware SFT methods (Lukasik et al., 2025; Chiang et al., 2025) across multiple benchmarks (See Fig. 3). Notably, REAL exhibits superior generalization to out-of-domain tasks, highlighting the importance of modeling regression structures directly within RL. While concurrent works like J1 (Whitehouse et al., 2025) also apply RL to the LLM-as-a-Judge setting, they fail to exploit the ordinal nature of numeric scoring. To the best of our knowledge, we are the first to successfully integrate a regression-aware objective directly into the RL training pipeline for LLM evaluators.

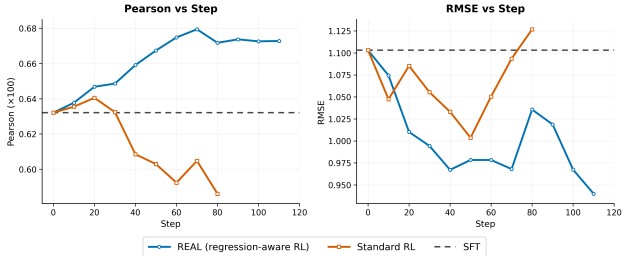

*Figure 2.* Evaluation performance during RL training. Both standard RL with binary reward (i.e., $r_{acc} = \mathbf{1}(y = y^*)$) and REAL with regression-aware reward (i.e., Eq. 5) were initialized from the SOTA SFT checkpoint, i.e., TRACT (Chiang et al., 2025). Standard RL results in suboptimal performance in correlation metrics compared to our proposed approach: REAL.

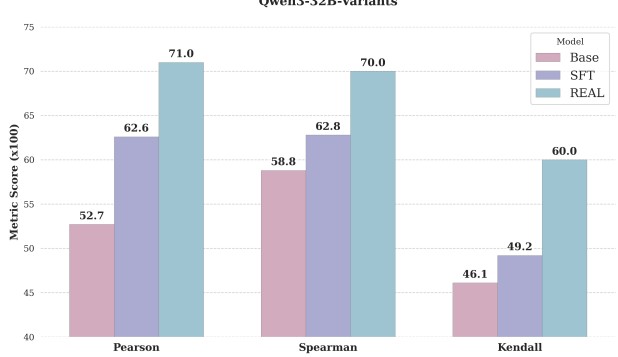

*Figure 3.* Performance gains on Qwen3-32B. Our method achieves an average increase of +8.40 Pearson and +7.20 Spearman correlation over the SFT baseline, and +18.30/+11.20 over the base model.

In summary, our contributions are threefold:

1. We propose REAL, a principled regression-aware RL framework that bridges the gap between standard RL and regression-based tasks by introducing a policy-dependent regression loss, as presented in Fig. 1. This formulation explicitly models the ordinal structure of evaluation tasks, addressing the inherent limitations of binary rewards (See Fig. 2). Furthermore, we provide a theoretical proof demonstrating that minimizing this regression loss also optimizes correlation metrics, as presented in Sec. 3.

2. We employ a generalized policy gradient estimator that enables mathematically sound optimization of policy-dependent objectives. This derivation reveals a functional decomposition that decouples the exploration of reasoning trajectories (CoT) from the optimization of final numerical predictions.

3. Experiments across model scales (8B to 32B) demonstrate that REAL consistently outperforms both

regression-aware SFT baselines and standard RL methods. REAL exhibits superior generalization on out-of-domain benchmarks; specifically, on Qwen3-32B, we achieve gains of +8.40 Pearson and +7.20 Spearman correlation over the SFT baseline, and +18.30/+11.20 over the base model, as shown in Fig. 3.

## 2. Background

### 2.1. LLMs and Reinforcement Learning

LLMs are typically trained in phases, starting from pre-training (Kaplan et al., 2020; Brown et al., 2020), following with multiple stages of post-training (Ouyang et al., 2022), most notably an alignment phase where an LLM is trained to improve its reasoning capabilities via Reinforcement Learning (RL) (Guo et al., 2025). Let $x$ denote the input prompt, $c$ the chain-of-thought (CoT), and $y$ the final answer. We define $\mathcal{D}$ as the training set, which, depending on the context, may consist of prompts $\{x\}$, prompt-target pairs $\{(x, y^*)\}$, or triplets $\{(x, c^*, y^*)\}$. Here, the superscript $^*$ denotes ground truth, as opposed to a predicted value from the model. Furthermore, let $\pi_\theta(\cdot \mid x)$ represent the distribution over token sequences, typically the CoT reasoning $c$ followed by the final answer $y$, as generated by a language model with learnable parameters $\theta$. Fundamentally, the objective of reinforcement learning (Sutton et al., 1999) is to maximize a reward:

$$\mathcal{L}(\theta) = \mathbb{E}_{x \sim \mathcal{D}, (c,y) \sim \pi_\theta(\cdot|x)} \left[ r(x, y) \right], \quad (1)$$

where the reward function $r$ is commonly an accuracy-based reward in the form of $\mathbf{1}(y = y^*)$ if the ground-truth label $y^*$ is available in the training dataset.

Since the reward $r(x, y)$ in LLM applications is typically only available after the entire sequence is generated, it is common to treat the full completion as a single action, resulting in a bandit-style formulation. This enables the use of the REINFORCE gradient estimator (Williams, 1992; Nguyen et al., 2017). Under this formulation, the policy gradient can be written as:

$$\nabla_\theta \mathcal{L}(\theta) = \mathbb{E}_{x \sim \mathcal{D}, (c,y) \sim \pi_\theta(\cdot|x)} \left[ r(x, y) \nabla_\theta \log \pi_\theta(y, c|x) \right]. \quad (2)$$

In this formulation, the reward $r(x, y)$ serves as a scalar weighting factor on the policy gradient, amplifying updates for high-reward generations while attenuating those associated with low rewards. We note that the derivation of the gradient relies on the assumption that the reward does not explicitly depend on the policy parameters, i.e., $\nabla_\theta r(x, y) = 0$.

### 2.2. Regression-Aware Inference and Finetuning in LLMs

Standard supervised fine-tuning (SFT) for LLMs typically relies on cross-entropy loss (Brown et al., 2020). This objective is suboptimal for regression tasks as it treats all incorrect numerical tokens equally, failing to account for their metric distance from the target value. To address this, several methods have been proposed to align LLM outputs with the inherent structure of numerical targets.

**Regression-Aware Fine-Tuning (RAFT)** (Lukasik et al., 2025) replaces token-level classification with a squared error loss. This loss is computed between the ground-truth score and an expected-value predictor, which applies a Bayes-optimal decision rule to the model's output distribution. Specifically, given an input $x$ and ground-truth reasoning $c^*$ and label $y^*$ from a dataset $\mathcal{D}$, the RAFT objective to *minimize* is defined as:

$$\mathcal{L}_{\text{RAFT}}(\theta) = \mathbb{E}_{(x, c^*, y^*) \sim \mathcal{D}} \\ [\underbrace{(\hat{y}_\theta(x, c^*) - y^*)^2}_{\text{regression loss}} - \underbrace{\lambda \log \pi_\theta(c^*, y^* | x)}_{\text{next token prediction loss}}], \quad (3)$$

where $\hat{y}_\theta(x, c) = \sum_{k \in \mathcal{K}} k \cdot \pi_\theta(k|x, c)$ denotes the expected value of the numerical prediction, originally proposed as the **RAIL predictor** (see Alg. 1) for regression-aware inference (Lukasik et al., 2024). Here, $\mathcal{K} = \{0, 1, \ldots, 9\}$ represents the set of single-digit numerical tokens, and $\lambda$ is a hyperparameter balancing the regression objective with the standard next-token prediction (NTP) loss. The RAIL predictor has been shown to offer "free-lunch" improvements in Pearson and Spearman correlation (Lukasik et al., 2025; Chiang et al., 2025), a finding we corroborate in Tab. 2.

---

**Algorithm 1** RAIL predictor (Lukasik et al., 2024)

---

**Require:** Input prompt $x$, LLM $\pi_\theta$
 1: Generate chain of thought $c \sim \pi_\theta(\cdot|x)$
    *# Compute expected value over digit token set $\mathcal{K}$*
 2: $\hat{y}_\theta \leftarrow \sum_{k \in \mathcal{K}} k \cdot \pi_\theta(k \mid x, c)$
 3: **return** RAIL predictor $\hat{y}_\theta$

---

Building on this foundation, **TRACT** (Chiang et al., 2025) extends the RAFT framework by incorporating self-generated reasoning paths. TRACT utilizes a two-stage SFT pipeline: initially, a temporary model $\pi_{\text{temp}}$ is trained via the RAFT objective (Eq. 3). Subsequently, a refined dataset is synthesized by pairing the original inputs $x$ with reasoning trajectories $c$ sampled from the temporary policy, $c \sim \pi_{\text{temp}}(\cdot|x)$, while retaining the ground-truth labels $y^*$. The final model is then trained on this augmented triplet dataset $\{(x, c, y^*)\}$ using the same RAFT objective. While this self-generation strategy significantly improves performance, it remains fundamentally a static SFT procedure. It

lacks a principled mechanism for continuous, active exploration of the reasoning space—a limitation we overcome by transitioning to a reinforcement learning framework.

## 3. Optimal Decision Rule for LLM-as-a-Judge Evaluation Metrics

Standard approaches for LLM post-training often frame reinforcement learning using a binary accuracy reward (i.e., the indicator function $\mathbf{1}\{y = y^*\}$), strictly optimizing for prediction correctness while performing inference via standard stochastic decoding. We argue that this paradigm is theoretically suboptimal for benchmarks primarily evaluated on correlation metrics, such as LLM-as-a-Judge (Gu et al., 2024; Chiang & Lee, 2023; Zheng et al., 2023b; Liu et al., 2023a; Zhu et al., 2024; Ouyang et al., 2022; Zhu et al., 2025). Prior works on regression-aware LLMs showed that other decision rules are optimal for regression loss (Lukasik et al., 2025; Chiang et al., 2025), but an optimal rule for correlation has not been discussed in these works, despite evaluating on correlation.

A fundamental challenge in optimizing for correlation is that it is inherently a population statistic defined over a set of predictions, rather than a single data point. Consequently, correlation cannot be directly utilized as a standard per-sample reward signal in reinforcement learning. We formally demonstrate that this limitation can be bypassed by identifying a shared optimal estimator: the squared error loss, a tractable per-sample objective, serves as the natural proxy for maximizing Pearson correlation.

**Lemma 3.1** (Optimality of Squared Error for Pearson Correlation). *Consider the following distributions: the distribution over input prompts $x \sim P_{\mathcal{D}}(.)$, the distribution over chains of thought (CoTs) $c \sim \pi_\theta(. \mid x)$ generated by the LLM policy $\pi_\theta$, and the distribution over targets conditioned on the inputs, $y^* \sim P(. \mid x)$. In this case, we assume the conditional independence $c \perp y^* \mid x$, yielding the joint distribution: $P_\theta(x, c, y^*) = P_{\mathcal{D}}(x)\pi_\theta(c \mid x)P(y^* \mid x)$. All expectations $\mathbb{E}[\cdot]$ below are taken with respect to $P_\theta(x, c, y^*)$. Define the posterior mean $\mu(x, c) \triangleq \mathbb{E}[y^* \mid x, c]$ and the squared-error risk $\mathcal{R}(\hat{y}) \triangleq \mathbb{E}\left[\left(\hat{y}(x, c) - y^*\right)^2\right]$. Assume $\mathrm{Var}(y^*) > 0$ and $\mathrm{Var}(\mu(x, c)) > 0$. The risk $\mathcal{R}(\hat{y})$ is minimized (a.s.) by $\hat{y}^*(x, c) = \mu(x, c)$, and the Pearson correlation $\rho(\hat{y}(x, c), y^*)$ is maximized by any positive affine transform of $\mu$: $\hat{y}(x, c) = a\,\mu(x, c) + b, \quad a > 0, \ b \in \mathbb{R}$. In particular, the squared-error minimizer $\hat{y}^*(x, c) = \mathbb{E}[y^* \mid x, c]$ is also optimal for maximizing Pearson correlation. In other words, by minimizing the squared error, we implicitly train the predictor to be the optimal estimator for the Pearson correlation metric.*

The proof, provided in Appendix E, establishes that the Pearson correlation is maximized by any positive linear transformation of the conditional expectation (Lemma E.1), and that the squared error loss is uniquely minimized by that same conditional expectation (Lemma E.2). By minimizing the squared loss, we implicitly train the predictor to be the optimal estimator for the Pearson correlation metric. This result aligns with empirical findings in RAFT (Lukasik et al., 2025) and TRACT (Chiang et al., 2025), which demonstrate that regression-aware objectives are effective for both squared error and correlation-based evaluation metrics.

## 4. REAL: Regression-Aware Reinforcement Learning

In this section, we introduce Regression-Aware Reinforcement Learning (REAL), a framework that leverages regression-based rewards to optimize LLM evaluators. As theoretically demonstrated in Sec. 3, this formulation is optimal for maximizing correlation metrics.

We first note that REAL extends the principles of regression-aware supervised fine-tuning—specifically RAFT (Lukasik et al., 2025) and TRACT (Chiang et al., 2025)—into the reinforcement learning regime. By transitioning to RL, the model is no longer confined to static, ground-truth reasoning but is instead able to explore and refine its own self-generated reasoning trajectories. A key distinction in our framework is that the resulting regression reward is policy-dependent, which necessitates the use of a generalized policy gradient to ensure valid optimization. This estimator naturally decomposes the training process into two distinct components: the exploration of reasoning pathways and the refinement of numerical predictions.

### 4.1. REAL Objective

We define the REAL objective to *minimize* $\mathcal{L}_{\text{REAL}}(\theta)$ as:

$$\mathcal{L}_{\text{REAL}}(\theta) = \mathbb{E}_{(x,y^*)\sim\mathcal{D},c\sim\pi_\theta(\cdot|x)} \\ \left[(\hat{y}_\theta(x, c) - y^*)^2 - \lambda \log \pi_\theta(y^*|x, c)\right], \quad (4)$$

In other words, we are augmenting the regression loss with an auxiliary log-likelihood term on the final answer tokens. When setting $\lambda = 0$ we recover the exact regression loss, which is of key importance for endowing the model with better numerical prediction abilities. Augmenting with the log-likelihood term is also in line with the SFT objective design in TRACT (Chiang et al., 2025).

Consequently, the implicit reward function in our framework is defined as:

$$r_{\text{REAL}}(\theta, x, c) = -(\hat{y}_\theta(x, c) - y^*)^2 + \lambda \log \pi_\theta(y^*|c, x). \quad (5)$$

Crucially, unlike standard RL post training where the reward is provided by a fixed external model, our reward function $r_{\text{REAL}}$ **explicitly depends on the policy parameters** $\theta$ (via

the estimator $\hat{y}_\theta$ and probability $\pi_\theta$).

## 4.2. Generalized Policy Gradient with Policy-Dependent Reward Functions

In standard LLM post-training methodologies, the reward function is typically modeled as an external signal derived from the environment or a fixed preference model, independent of the current policy $\pi_\theta$ (i.e., $\nabla_\theta r = 0$). However, our framework introduces a reward function $r_{\text{REAL}}(\theta, x, c)$ (Eq.5) that explicitly depends on the policy parameters $\theta$. Consequently, the standard policy gradient formulation is insufficient. In this section, we formalize the generalized policy gradient (Schulman et al., 2015) for this class of policy-dependent reward functions.

**Lemma 4.1** (Generalized Policy Gradient with Policy-Dependent Rewards for Regression). *Let the objective be* $\mathcal{L}(\theta) = \mathbb{E}_{x\sim\mathcal{D}}\mathbb{E}_{c\sim\pi_\theta(\cdot|x)}[r(\theta, x, c)]$. *The gradient* $\nabla_\theta\mathcal{L}(\theta)$ *is given by:*

$$\nabla_\theta\mathcal{L}(\theta) = \mathbb{E}_{(x,y^*)\sim\mathcal{D},c\sim\pi_\theta(\cdot|x)}$$
$$\left[\underbrace{r(\theta, x, c)\nabla_\theta\log\pi_\theta(c\mid x)}_{\text{Term 1: CoT Update}} + \underbrace{\nabla_\theta r(\theta, x, c)}_{\text{Term 2: Prediction Update}}\right].$$
$$(6)$$

*The first term optimizes the CoT generation, weighted by the reward; the second term provides supervision for final answer prediction through standard backpropagation. Specifically, by substituting the REAL reward function (Eq. 5) into the general gradient expression (Eq. 6), we obtain the gradient for the REAL objective (Eq. 4). Specifically, Term 2 can be expanded as:*

$$\text{Term 2} = -2(\hat{y}_\theta(x, c) - y^*)\nabla_\theta\hat{y}_\theta(x, c)$$
$$+ \lambda\nabla_\theta\log\pi_\theta(y^*|x, c) \quad (7)$$

*where the gradient of the predicted value is* $\nabla_\theta\hat{y}_\theta(x, c) = \sum_{k\in\mathcal{K}} k \cdot \nabla_\theta\pi_\theta(k|x, c)$.

**Distinguishing Reasoning Exploration from Prediction Refinement** The REAL gradient (i.e., Eq. 6) explicitly decomposes optimization into two distinct parts: exploring reasoning trajectories (Term 1) and refining score prediction (Term 2). This separation is critical because standard RL approaches like GRPO (Shao et al., 2024) treat the reasoning chain $c$ and final answer $y$ as a homogeneous sequence, applying a uniform update rule. In contrast, our framework addresses their structural differences: the CoT update allows exploration via policy gradients, whereas the final score targets a known ground truth, allowing for direct correction via standard backpropagation.

## 4.3. Stabilization and Implementation

To enhance the stability of the REAL objective during training, we utilize the RLOO estimator (Kool et al., 2019) for the first term, which governs reasoning exploration. Other stabilization approaches (Schulman et al., 2017; Guo et al., 2025) can be explored in future work. We note that this is not central to the core contribution of our paper, i.e., the regression-aware RL objective and algorithm.

For a given input $(x, y^*)$, we sample $K$ independent CoT trajectories $\{c_1, c_2, \ldots, c_K\}$ from the current policy $\pi_\theta(\cdot|x)$. Let $r^{(i)} = r_{\text{REAL}}(\theta, x, c_i)$ denote the reward associated with the $i$-th trajectory. The advantage $A^{(i)}$ is computed as: $A^{(i)} = r^{(i)} - b^{(i)}; b^{(i)} = \frac{1}{K-1}\sum_{j\neq i} r^{(j)}$. We further compute the standardized advantage $\tilde{A}^{(i)} = \text{clip}\left(\frac{A^{(i)}}{\sigma(A)+\epsilon}, -1, 1\right)$ for better training stability, where $\sigma(A)$ denotes the standard deviation of the advantages within the sampled group and $\epsilon = 10^{-8}$. Then from Eq. 6, replacing reward by RLOO advantages in Term 1 and add $\beta$ on Term 2 controls the strength of the prediction updates, we have the stabilized gradient estimator defined as

$$\nabla_\theta\mathcal{L}_{\text{REAL}}^{\text{RLOO}}(\theta) \approx \frac{1}{K}\sum_{i=1}^{K}\left[\underbrace{\tilde{A}_{\text{REAL}}^{(i)}\nabla_\theta\log\pi_\theta(c_i|x)}_{\text{CoT Update}} + \beta\underbrace{\nabla_\theta r_{\text{REAL}}(\theta, x, c_i)}_{\text{Prediction Update}}\right]$$
$$(8)$$

Following (Tang et al., 2025) we introduce a prediction weight $\beta$ to balance Term 1 and Term 2. Note that Eq. 6 implies an exact weighting of $\beta = 1.0$, which we find to already bring good results, as shown in Tab. 9.

## 4.4. Relations with Prior Methods

We provide a complete conceptual comparison with prior methods in Tab. 1.

**Comparison to TRACT: Active vs. Static Reasoning** We identify both a conceptual connection and a critical limitation in TRACT (Chiang et al., 2025). While TRACT introduces reasoning traces to the regression setting, it remains confined to the SFT paradigm. Specifically, TRACT treats all self-generated Chain-of-Thought (CoT) trajectories as ground truth, failing to evaluate their intermediate quality. In contrast, REAL leverages a regression-aware reward to actively explore and rank CoT trajectories during RL. Mathematically, TRACT's update is equivalent to the *prediction refinement* term (Term 2) of our objective, but it lacks the *trajectory exploration* component (Term 1) necessary for true reinforcement learning, as can be seen when comparing the respective rows in Tab. 1. Thus, TRACT can be viewed as an offline, supervised subset of the more general REAL framework.

**Regression-Aware vs. Standard RL** Standard RL typically relies on coarse binary rewards (e.g., 0/1 accuracy), which ignores the ordinal structure inherent in evaluation tasks. As illustrated in Fig. 2, REAL utilizes the policy's full probability distribution to shape a fine-grained, regression-

*Table 1.* Conceptual comparison of selected SFT and RL training methods on non-ordinal and ordinal targets. $r(\theta, x, c) = -(\hat{y}_\theta(x, c) - y^*)^2 + \lambda \log \pi_\theta(y^*|x, c)$ represents the regression-aware reward. REAL is the first framework to combine a regression-aware reward with reinforcement learning for reasoning exploration. Note that JEPO further approximates the gradient with the multi-sample Jensen's Bound.

| Method | Paradigm | CoT Exploration | Ordinal Awareness | Gradient |
|---|---|---|---|---|
| SFT | SFT | ✗ | ✗ | $\mathbb{E}_{(x, y^*) \sim \mathcal{D}} [\nabla_\theta \log \pi_\theta(y^*|x)]$ |
| RAFT/TRACT | SFT | ✗ | ✓ | $\mathbb{E}_{(x, c, y^*) \sim \mathcal{D}} [\nabla_\theta r(\theta, x, c)]$ |
| Standard RL | RL | ✓ | ✗ | $\mathbb{E}_{x \sim \mathcal{D}, (c, y) \sim \pi_\theta(\cdot|x)} [r(x, y) \nabla_\theta \log \pi_\theta(y, c|x)]$ |
| JEPO (Tang et al., 2025) | RL | ✓ | ✗ | $\mathbb{E}_{x \sim \mathcal{D}, c \sim \pi_\theta(\cdot|x)} [\log \pi_\theta(y^*|x, c) \nabla_\theta \log \pi_\theta(c|x) + \nabla_\theta \log \pi_\theta(y^*|x, c)]$ |
| REAL (ours) | RL | ✓ | ✓ | $\mathbb{E}_{(x, y^*) \sim \mathcal{D}, c \sim \pi_\theta(\cdot|x)} [r(\theta, x, c) \nabla_\theta \log \pi_\theta(c \mid x) + \nabla_\theta r(\theta, x, c)]$ |

aware reward. This allows the model to distinguish "near-miss" predictions from total failures, leading to significantly higher alignment with human judgment as demonstrated by the correlation metrics in Tab. 2.

**Comparison to JEPO:** JEPO (Tang et al., 2025) optimizes the marginal log-likelihood $\log p(y^*|x)$ using Jensen's bound to handle unverifiable data. While effective for tasks like mathematical proofs, JEPO lacks the ordinal awareness required for numeric scoring in LLM-as-a-Judge settings. Our REAL objective (4) is explicitly regression-aware, providing a more principled approach for tasks where the distance between scores matters. We contrast JEPO and our objectives in Tab. 1. Since JEPO does not release an official codebase, we reproduced their results in our setting. The empirical comparisons in Tab. 14 in the Appendix confirm that REAL consistently outperforms JEPO across all regression metrics.

## 5. Experiments

We mainly follow the experimental setup in (Chiang et al., 2025; Kim et al., 2023; 2024) for evaluation, and implement our RL algorithm using the codebase of verl (Sheng et al., 2024).

**Models** We implement REAL on three LLMs with different architectures, *tokenizers*, and sizes to demonstrate robustness across backbones: 1) **Mistral2-7B** (Jiang et al., 2023), for a fair comparison with TRACT and Prometheus-2, which both use it as their base model; 2) **Qwen3-8B** (Yang et al., 2025), which uses a different tokenizer than Mistral and has recently shown strong reasoning and instruction-following abilities; and 3) **Qwen3-32B** (Yang et al., 2025), to which we scale up performance to demonstrate the scalability of REAL.

**Datasets** We implement training on **Feedback Collection** (Kim et al., 2023), which consists of approximately 100K pointwise samples together with 1K fine-grained score rubrics, 20K instructions. Following (Chiang et al., 2025; Kim et al., 2023; 2024), we evaluate pointwise LLM-as-

a-Judge performance on four widely used benchmarks: 1) **Feedback Bench**, the official test set for Feedback Collection, includes 1K *non-overlapping* score rubrics and 200 instructions; 2) **FLASK**, a fine-grained evaluation benchmark, consists of 200 prompts, 12 score rubrics, and 2K responses generated by Alpaca-7B (Taori et al., 2023), Vicuna-13B (Chiang et al., 2023), Bard (Google, 2023), and GPT-3.5-Turbo-0613 (OpenAI, 2023); 3) **Vicuna Bench**, a single-turn dialogue dataset, includes 80 test prompts, 80 hand-crafted score rubrics, and 320 responses generated by WizardLM-13B (Xu et al., 2023), Vicuna-13B (Chiang et al., 2023), Llama-2Chat-13B (Touvron et al., 2023), and GPT-3.5-Turbo-0613 (OpenAI, 2023); and 4) **MT Bench**, a multi-turn chat benchmark, contains 80 test prompts, 80 hand-crafted score rubrics, and 320 responses generated by the same models as above. Together, these datasets cover a diverse range of prompts, score rubrics, response styles, and difficulty levels. In addition, we note that Feedback Bench can be regarded as more of an *in-domain* test dataset, while the other three are *out-of-distribution*.

**Evaluation Metrics** We report the Pearson correlation coefficient $r$ (Pearson, 1895), Spearman's rank correlation $\rho$ (Spearman, 1961), and Kendall's $\tau$ (Kendall, 1938) with ground-truth scores. Additionally, we provide RMSE and MAE metrics in the ablation study for reference.

**Baselines** We compare REAL against zero-shot, SFT, and RL baselines:

(i) **Zero-shot** inference using the base model;

(ii) **RAFT** (Lukasik et al., 2025), which applies regression-aware supervised fine-tuning with Eq. 3;

(iii) **TRACT** (Chiang et al., 2025), which performs two-stage regression-aware supervised fine-tuning. Note that TRACT simply treats all self-generated CoTs as ground truth for the second stage, without any CoT update. In contrast, our method conducts CoT policy gradient updates weighted by the REAL reward;

*Table 2.* **Evaluation results (×100) across benchmarks using Pearson ($r$), Spearman ($\rho$), and Kendall ($\tau$).** All baseline reward model scores are taken from TRACT (Chiang et al., 2025) and Prometheus (Kim et al., 2023). Results for the Mistral and Qwen models are obtained from our own runs. Underline indicates the initialization checkpoint used by REAL. We find that TRACT saturates on the Qwen3 series models, so we choose RAFT as the initialization checkpoint. Mistral2-7B's base model failed to follow instructions and respond in the correct format, so we applied a 100-step warmup. Due to computational constraints we run only RAFT and REAL for Qwen3-32B; we include all baselines for other models.

| Method | Training | Inference | FB Bench | | | FLASK | | | Vic. Bench | | | MT Bench | | | Average | | |
|---|---|---|---|---|---|---|---|---|---|---|---|---|---|---|---|---|---|
| | | | $r$ | $\rho$ | $\tau$ | $r$ | $\rho$ | $\tau$ | $r$ | $\rho$ | $\tau$ | $r$ | $\rho$ | $\tau$ | $r$ | $\rho$ | $\tau$ |
| | | | *Baseline Reward Models* | | | | | | | | | | | | | | |
| GPT-3.5-Turbo-0613 | None | Standard | 56.3 | 52.1 | 45.3 | 27.0 | 23.2 | 18.7 | 27.5 | 26.7 | 20.2 | 42.2 | 37.1 | 29.9 | 38.3 | 34.8 | 28.5 |
| Prometheus-1-7B | SFT | Standard | 84.7 | 84.9 | 76.7 | 45.7 | 45.7 | 36.5 | 29.3 | 29.5 | 21.6 | 36.7 | 37.1 | 28.5 | 49.1 | 49.3 | 40.8 |
| Prometheus-1-13B | SFT | Standard | **86.0** | **85.8** | **77.1** | 46.6 | 42.9 | 34.6 | 47.3 | 45.1 | 34.1 | 46.7 | 45.5 | 34.5 | 56.7 | 54.8 | 45.1 |
| Prometheus-2-7B | SFT | Standard | 84.5 | 84.7 | 76.5 | 51.2 | 49.3 | **40.5** | 48.8 | 48.0 | **41.1** | 51.9 | 48.3 | **39.2** | 59.1 | 57.6 | **49.3** |
| Prometheus-2-7B | SFT | RAIL | 85.3 | 85.3 | 72.9 | **52.5** | **51.4** | 39.2 | **51.0** | **51.3** | 40.3 | **53.8** | **51.1** | 38.9 | **60.7** | **59.8** | 47.8 |
| | | | *Mistral2-7B-Variants* | | | | | | | | | | | | | | |
| Base (w/ warmup) | None | Standard | 83.1 | 83.3 | 74.8 | 41.5 | 41.9 | 34.1 | 49.2 | 42.4 | 36.1 | 30.9 | 31.8 | 25.3 | 51.2 | 49.8 | 42.6 |
| Base (w/ warmup) | None | RAIL | 83.7 | 84.3 | 70.2 | 42.5 | 43.7 | 32.3 | 50.3 | 45.4 | 34.0 | 32.0 | 29.9 | 21.4 | 52.1 | 50.8 | 39.5 |
| RAFT | SFT | RAIL | 87.9 | 88.0 | 76.3 | 41.8 | 41.9 | 31.5 | 52.8 | 51.3 | 40.9 | 39.9 | 41.8 | 30.7 | 55.6 | 55.8 | 44.8 |
| TRACT | SFT | RAIL | **93.9** | **93.7** | **82.9** | 50.7 | 50.0 | 37.2 | 56.2 | 54.8 | 42.6 | 52.1 | 50.1 | 36.6 | 63.2 | 62.2 | 49.8 |
| Standard RL | RL | RAIL | 93.7 | 93.7 | 82.8 | 51.6 | 50.5 | 37.9 | 58.0 | 56.0 | 43.4 | 52.9 | 50.7 | 37.1 | 64.1 | 62.7 | 50.3 |
| REAL (ours) | RL | RAIL | 93.2 | 93.4 | 82.5 | **56.0** | **54.1** | **41.1** | **63.3** | **60.2** | **46.3** | 59.3 | 56.9 | 42.2 | **67.9** | **66.2** | **53.0** |
| | | | *Qwen3-8B-Variants* | | | | | | | | | | | | | | |
| Base | None | Standard | 56.6 | 62.7 | 53.9 | 44.8 | 48.0 | 40.1 | 37.3 | 46.5 | 40.8 | 35.5 | 32.7 | 26.5 | 43.6 | 47.4 | 40.3 |
| Base | None | RAIL | 56.7 | 65.4 | 54.1 | 45.0 | 48.3 | 38.5 | 37.3 | 46.1 | 38.6 | 35.9 | 35.5 | 27.0 | 43.7 | 48.8 | 39.5 |
| RAFT | SFT | RAIL | 84.3 | 85.5 | 73.0 | 49.2 | 50.1 | 37.5 | 59.9 | 57.2 | 44.0 | 54.1 | 51.7 | 38.5 | 61.9 | 61.1 | 48.3 |
| TRACT | SFT | RAIL | **94.9** | **94.7** | 84.3 | 51.4 | 51.3 | 38.5 | 50.5 | 50.4 | 38.3 | 55.8 | 58.6 | 43.6 | 63.1 | 63.8 | 51.2 |
| Standard RL | RL | RAIL | 94.0 | 94.2 | 83.5 | 52.3 | 52.4 | 39.5 | 57.1 | 56.1 | 43.0 | 49.5 | 53.5 | 39.7 | 63.2 | 64.0 | 51.4 |
| REAL (ours) | RL | RAIL | 92.0 | 92.1 | **85.7** | **53.8** | **53.9** | **43.1** | **60.5** | **57.8** | **48.2** | 61.7 | 60.8 | 47.1 | **67.0** | **66.0** | **56.1** |
| | | | *Qwen3-32B-Variants* | | | | | | | | | | | | | | |
| Base | None | RAIL | 63.4 | 70.8 | 56.7 | 54.3 | **60.4** | 47.2 | 50.8 | 57.4 | 45.0 | 42.5 | 46.8 | 35.3 | 52.7 | 58.8 | 46.1 |
| RAFT | SFT | RAIL | 85.4 | 86.5 | 72.9 | 52.1 | 52.9 | 39.9 | 51.9 | 52.0 | 39.9 | 61.1 | 59.6 | 43.9 | 62.6 | 62.8 | 49.2 |
| REAL (ours) | RL | RAIL | **91.1** | **91.7** | **85.9** | **58.9** | 58.6 | **47.4** | **65.1** | **60.7** | **51.2** | 68.9 | 69.1 | 55.2 | **71.0** | **70.0** | **60.0** |

*Table 3.* **Average-of-N results at inference.** Averaging the results of $N$ generations slightly improves performance, although REAL already enjoys efficiency and strong performance at $N = 1$.

| Setting | $r \uparrow$ | $\rho \uparrow$ | $\tau \uparrow$ | RMSE $\downarrow$ | MAE $\downarrow$ |
|---|---|---|---|---|---|
| $N = 1$ | 67.9 | 66.2 | 53.0 | 0.968 | 0.697 |
| $N = 4$ | 68.1 | 66.2 | 53.0 | 0.965 | 0.695 |
| $N = 10$ | **68.1** | 66.2 | 53.0 | **0.964** | **0.695** |

(iv) **Standard RL (accuracy reward)** (Kool et al., 2019), a baseline that uses accuracy as the reward, where $r_{acc} = \mathbf{1}(y = y^*)$, and updates via $A_{acc} \nabla_\theta \pi(c, y \mid x)$, where $A_{acc}$ is the RLOO advantage for $r_{acc}$;

(v) **Other baseline reward models:** 1) zero-shot GPT3.5-Turbo (OpenAI, 2023); 2) Prometheus-1-7B (Kim et al., 2023); 3) Prometheus-1-13B (Kim et al., 2023); and 4) Prometheus-2-7B (Kim et al., 2024).

Following the findings in (Chiang et al., 2025; Lukasik et al., 2024), we mainly report RAIL inference results, as RAIL has been shown to offer *free-lunch* improvements under Pearson and Spearman correlation metrics at inference.

**Configuration** We conduct experiments using 8 NVIDIA A100 GPUs for Mistral2-7B and Qwen3-8B, and 2×8 A100

*Table 4.* **Comparison between standard RL baselines and REAL on Mistral2-7B.** Standard RL(reg) with regression-aware policy-independent rewards also fail to improve upon the SFT checkpoint.

| Method | Type | $r$ | $\rho$ | $\tau$ |
|---|---|---|---|---|
| TRACT | SFT | 63.2 | 62.2 | 49.8 |
| Standard RL(acc) | RL | 64.1 | 62.7 | 50.3 |
| Standard RL(reg) | RL | 63.4 | 62.6 | 50.1 |
| REAL | RL | **67.9** | **66.2** | **53.0** |

*Table 5.* **Ablation on group size $K$.** Larger values of $K$ generally result in better correlation performance.

| Setting | $r \uparrow$ | $\rho \uparrow$ | $\tau \uparrow$ | RMSE $\downarrow$ | MAE $\downarrow$ |
|---|---|---|---|---|---|
| $K = 4$ | 63.7 | 62.6 | 50.2 | 1.063 | 0.775 |
| $K = 8$ | 67.2 | 65.0 | 52.0 | **0.943** | **0.679** |
| $K = 16$ | **67.9** | **66.2** | **53.0** | 0.968 | 0.697 |

GPUs for Qwen3-32B. All experiments are performed using full-parameter fine-tuning. The learning rate is set to $5 \times 10^{-8}$ for the Mistral models and $1 \times 10^{-6}$ for the Qwen series. The *max_response_length* is set to 1024. Unless otherwise specified, other hyperparameters are shared across all three evaluated LLMs. Ablation studies are carried out

*Table 6.* **Ablation on log-likelihood weight** $\lambda$. Using regression-aware reward alone ($\lambda = 0$) already yields excellent results, while extra log-likelihood reward further improves.

| Setting | $r \uparrow$ | $\rho \uparrow$ | $\tau \uparrow$ | RMSE $\downarrow$ | MAE $\downarrow$ |
|---|---|---|---|---|---|
| TRACT | 63.2 | 62.2 | 49.8 | 1.103 | 0.813 |
| $\lambda = 0$ | 66.0 | 64.6 | 51.8 | 1.027 | 0.736 |
| $\lambda = 1$ | 67.9 | **66.2** | **53.0** | 0.968 | **0.697** |
| $\lambda = 5$ | **68.1** | 65.8 | 52.7 | 0.955 | 0.719 |
| $\lambda = 10$ | **68.1** | 66.1 | 52.9 | **0.928** | 0.699 |

*Table 7.* **Ablation on dynamic sampling when group size** $K = 8$. Partially correct prompts provide the best learning signal.

| Setting | $r \uparrow$ | $\rho \uparrow$ | $\tau \uparrow$ | RMSE $\downarrow$ | MAE $\downarrow$ |
|---|---|---|---|---|---|
| $0 < \text{acc} < 1$ | **67.2** | **65.0** | **52.0** | **0.943** | **0.679** |
| $0 < \text{acc} \leq 1$ | 65.6 | 64.1 | 51.3 | 1.000 | 0.711 |
| $0 \leq \text{acc} < 1$ | 65.3 | 63.4 | 50.9 | 1.001 | 0.728 |
| $0 \leq \text{acc} \leq 1$ | 65.5 | 63.9 | 51.1 | 0.990 | 0.715 |

to analyze the impact of different hyperparameter settings on model performance.

### 5.1. Main Results

See Tab. 2 for all the results. Across all benchmarks and model backbones, REAL consistently improves correlation with ground-truth scores compared to the zero-shot inference, regression-aware supervised fine-tuning, and the standard RL methods. We note that REAL performs significantly better on *out-of-domain* benchmarks such as Vicuna, MT, and Flask. We provide a qualitative example in Appendix G.

**Average-of-N results at inference** We average the RAIL prediction $\hat{y}_\theta$ over $N$ generations for each test prompt as the final output and then calculate the metrics shown in Tab. 3. Scaling $N$ to 4 or 10 slightly improves performance, which indicates that REAL already achieves strong results at $N = 1$. This demonstrates that the model learns a robust policy capable of efficiently identifying high-reward outputs. This single-generation capability ensures peak sampling efficiency, making REAL highly suitable for low-latency, real-world deployment by eliminating the computational overhead of multi-sample averaging.

### 5.2. On Regression-Aware Policy-Independent Rewards

A natural question is whether the gains of REAL mainly come from the regression-aware reward itself, rather than from our generalized policy gradient formulation. To investigate this, we additionally evaluate a policy-independent regression reward under standard RL training. Specifically, Standard RL above uses an accuracy accuracy reward $r_{\text{acc}} = \mathbf{1}(y = y^*)$. We further construct a regression-

*Table 8.* **Initialization from different checkpoints.** Initializing from a high-quality SFT checkpoint can lead to better SOTA performance. Our algorithm works with different checkpoint types.

| Setting | $r \uparrow$ | $\rho \uparrow$ | $\tau \uparrow$ | RMSE $\downarrow$ | MAE $\downarrow$ |
|---|---|---|---|---|---|
| base | 52.1 | 50.8 | 39.5 | 1.225 | 0.901 |
| init. from base | **58.8** | **56.8** | **45.1** | **1.155** | **0.793** |
| $\Delta$ | +6.7 | +6.0 | +5.6 | -0.070 | -0.108 |
| SFT | 63.2 | 62.2 | 49.8 | 1.103 | 0.813 |
| init. from SFT | **67.9** | **66.2** | **53.0** | **0.968** | **0.697** |
| $\Delta$ | +4.7 | +4.0 | +3.2 | -0.135 | -0.116 |

*Table 9.* **Ablation on prediction weight** $\beta$. We choose $\beta = 0.01$ (See Eq. 8) for initialization from both base and SFT checkpoints.

| Setting | $r \uparrow$ | $\rho \uparrow$ | $\tau \uparrow$ | RMSE $\downarrow$ | MAE $\downarrow$ |
|---|---|---|---|---|---|
| | | *Init. from Base* | | | |
| $\beta = 0$ | 57.7 | **56.9** | **45.1** | 1.200 | 0.831 |
| $\beta = 0.01$ | **58.8** | 56.8 | **45.1** | **1.155** | **0.793** |
| $\beta = 1$ | 57.9 | 56.4 | 44.4 | 1.184 | 0.822 |
| | | *Init. from SFT* | | | |
| $\beta = 0$ | 67.3 | 65.9 | 52.5 | 0.980 | 0.701 |
| $\beta = 0.01$ | **67.9** | **66.2** | **53.0** | 0.968 | 0.697 |
| $\beta = 1$ | 66.7 | 64.5 | 51.6 | **0.935** | **0.677** |

aware but policy-independent reward using squared error, $r_{\text{reg}} = -(y - y^*)^2$ where $y = \arg\max_k \pi_\theta(k|x, c)$, and train it with standard RL, denoted as **Standard RL(reg)**. Tab. 4 shows that Standard RL(reg) behaves very similarly to Standard RL and quickly collapses during RL training, whereas REAL consistently improves performance.

**Intuitively, the key issue is that standard RL methods fail to evaluate the true quality of sampled trajectories.** For example, suppose the ground-truth score is $y^* = 5$. Since our REAL model is initialized from a strong SFT checkpoint, many sampled trajectories already correctly decode to $y = 5$. Under standard RL methods such as RLOO, PPO, or GRPO, all such trajectories receive identical rewards and advantages, regardless of whether $\pi_\theta(k = 5|x, c) = 0.9$ or $0.8$. Even with the regression-aware reward $r_{\text{reg}}$, the advantage remains identical as long as the decoded prediction $y$ is unchanged.

REAL circumvents this limitation by defining the reward on the predictive distribution, which explicitly depends on the policy parameters. Consequently, two trajectories that both decode to 5 may still yield different rewards (e.g., 4.99 vs. 4.70), providing a substantially richer learning signal. These results suggest that the gains of REAL do not simply come from replacing classification rewards with regression-style rewards. Instead, the key ingredient is combining a policy-dependent regression-aware reward with the generalized policy gradient required to optimize it.

*Table 10.* **Ablation on temperature** $T$. $T = 1$ is chosen by default. The temperature parameter is applied in both the training and inference stages, for both CoT generation and score prediction.

| Setting | $r\uparrow$ | $\rho\uparrow$ | $\tau\uparrow$ | RMSE$\downarrow$ | MAE$\downarrow$ |
|---------|------|------|------|-------|-------|
| $T = 0.8$ | 66.8 | 64.7 | 51.9 | 0.973 | 0.725 |
| $T = 1$ | **67.9** | **66.2** | **53.0** | **0.968** | **0.697** |

## 5.3. Ablation Studies

Rows shaded in gray indicate the default setting used in our experiments.

**Group size $K$: RLOO variance reduction results in superior performance** Tab. 5 indicates a clear positive correlation between the number of sampled responses and the quality of the learned model. Increasing the group size from $K = 4$ to $K = 16$ yields consistent improvements across all correlation metrics, with the Pearson correlation $r$ rising from 63.7 to 67.9, indicating the effectiveness of RLOO variance reduction.

**Log-likelihood weight $\lambda$: regression-aware reward alone excels in performance; additional log-likelihood term gives further benefits.** Tab. 6 shows that using the regression-aware reward alone ($\lambda = 0$) already significantly outperforms the TRACT baseline. Moreover, introducing the log-likelihood objective ($\lambda = 1/5/10$) provides further enhancement. This suggests that the log-likelihood component serves as a beneficial complement to the core regression objective. $\lambda = 1$ is chosen by default.

**Dynamic Sampling Strategy: Partially Correct Prompts Provide the Most Informative Learning Signal.** As shown in Tab. 7, we evaluate four training settings based on different dynamic sampling (Yu et al., 2025; Xu et al., 2025) strategies using *group accuracy* as the primary metric.

Let $\{y^{(k)}\}_{k=1}^{K}$ denote a group of $K$ generated responses for a given prompt $x$, and let $y^*$ be the ground-truth label. We define the group accuracy $\mathrm{acc}(x)$ as the proportion of correct responses within the group: $\mathrm{acc}(x) = \frac{1}{K}\sum_{k=1}^{K}\mathbf{1}\{y^{(k)} = y^*\}$. Aligned with (Yu et al., 2025), our results demonstrate that training exclusively on the *partially correct* set ($0 < \mathrm{acc} < 1$) consistently yields the best performance. This indicates that these prompts provide a rich learning signal while avoiding the gradient noise of entirely incorrect examples ($\mathrm{acc} = 0$) and the vanishing supervision of already mastered ones ($\mathrm{acc} = 1$).

**Initialization checkpoints, Prediction weight $\beta$, and Temperature $T$.** Tab. 8 demonstrates that REAL is robust to different starting configurations, yielding consistent improvements whether initialized from a base model or a high-quality SFT checkpoint. Tab. 9 indicates $\beta = 0.01$ brings the best performance regardless of initialization checkpoint

types. $T = 1$ is chosen by default as supported by Tab. 10.

## 5.4. Generalizability to Other Settings

**Extension to continuous-valued scores.** REAL supports continuous-valued prediction during both training and inference. Specifically, REAL outputs a continuous regression prediction $\hat{y}_\theta$, and the ground-truth labels $y^*$ can be arbitrary real values without needing to be integers. Our experiments include continuous-valued benchmarks such as FLASK, MT-Bench, and Vicuna, where the ground-truth targets are averages of multiple integer ratings from humans or GPT-4.

**Extension to wider score ranges.** For datasets with score ranges exceeding our default setting (e.g. $[1, 100]$), labels can simply be normalized into a smaller range during training, such as $[0, 9]$. This scales the targets to match the single-digit token space used by the expected-value predictor, which can then be mapped back through de-normalization at inference time.

**Extension to pairwise difference modeling.** REAL is applicable to pairwise scoring settings as well. For example, the HuggingFace `prometheus-eval/Preference_Collection` dataset provides scores for pairs of responses, denoting the preference of the first response over the second response. In case such scores span a wide range (e.g. negative values), they can be easily shifted into a non-negative range, after which REAL can be trained to predict the shifted difference score directly, as shown in Appendix C.

## 6. Limitations

REAL relies on the quality and calibration of the underlying score supervision. If training labels are noisy, inconsistent, or systematically biased, the regression-aware objective may optimize toward imperfect numerical targets. In addition, REAL relies on self-generated Chain-of-Thought reasoning during evaluation, which may inherit biases or systematic errors from the underlying base models.

## 7. Conclusion

We introduced REAL, a regression-aware reinforcement learning framework that employs the generalized policy gradient methods to solve regression-aware objectives. Our practical implementation combines RLOO stabilization with regression-aware prediction updates. Empirically, REAL achieves significant and consistent improvements across multiple LLM-as-a-Judge benchmarks over zero-shot and SFT baselines, specifically for out-of-domain evaluation. We believe this work provides a foundation for future research on RL algorithms for LLM-as-a-Judge.

## Acknowledgements

The authors would like to thank Junfeng He and Avinab Saha from Google for bringing us together. The authors would also like to thank the three anonymous reviewers for their constructive feedback. Ying Nian Wu was supported in part by NSF DMS-2415226, DARPA W912CG25CA007, and research gifts from Amazon and Qualcomm. Tianyu Chen and Mingyuan Zhou acknowledge the support of NSF-IIS 2212418 and NIH-R37 CA271186.

## Impact Statement

This work contributes to the development of more faithful regression-based LLM-as-a-Judge systems, which can improve the reliability of automatic evaluation for large-scale language model research. By better respecting the ordinal and numerical structure of evaluation scores, REAL has the potential to reduce noise, instability, and miscalibration in benchmark-driven model comparison, thereby supporting more reproducible and efficient research workflows. However, the use of automated evaluators raises important considerations regarding appropriate deployment. Although REAL improves numerical consistency and robustness, its predictions remain inherently model-dependent and reflect biases present in the underlying language model and training data. Consequently, such systems should not be interpreted as objective or normative arbiters of quality, particularly in high-stakes or value-laden decision-making contexts. Over-reliance on automated judgments may create a false sense of precision or neutrality when evaluation criteria are underspecified or subjective.

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

# A. Related Work

**Reinforcement Learning in LLMs**   Reinforcement learning (Sutton et al., 1999) has emerged as a key technique for fine-tuning large language models (LLMs), initially aimed at aligning model outputs with human preferences (Ouyang et al., 2022). In this setting, the RL formulation typically involves a single-generation, single-action structure (Nguyen et al., 2017). Popular approaches include REINFORCE (Williams, 1992) and Proximal Policy Optimization (PPO) (Schulman et al., 2017). RLOO (Kool et al., 2019) is an unbiased policy gradient estimator that reduces variance without requiring a learned value function. It improves upon standard REINFORCE by using a leave-one-out baseline, where each sample's advantage is computed relative to the average reward of the other samples in the batch. Beyond RLOO, actor–critic methods such as Proximal Policy Optimization (PPO) (Schulman et al., 2017) leverages intermediate state value functions to reduce gradient variance, at the cost of introducing *bias* into the estimator. Unlike RLOO, PPO typically requires an additional critic model, which increases computational overhead. During training, the updates of the generative policy and the critic are interleaved. Group Relative Policy Optimization (GRPO) (Guo et al., 2025) further removes the need for an explicit critic by normalizing rewards within a group of samples generated for the same prompt. While early RLHF methods focused on preference alignment, recent efforts have used reinforcement learning to explicitly incentivize desirable capabilities, such as step-by-step reasoning (Guo et al., 2025). This shift reflects a broader trend toward leveraging reward signals not only for alignment but also for improving the utility and robustness of LLMs in complex tasks. Recent studies (Ahmadian et al., 2024) have shown that RLOO can outperform biased actor-critic methods such as PPO on key baselines in large-scale settings. While we mainly use RLOO variance reduction techniques, other stabilization approaches can be explored in future work. We note that this is not central to the core contribution of our paper, i.e., the regression-aware RL objective and algorithm.

**Regression-Aware LLMs.**   Standard LLM fine-tuning uses a cross-entropy loss (Brown et al., 2020), which is suboptimal for regression tasks as it treats all incorrect numerical tokens equally, regardless of their numeric distance from the target. Several methods have been proposed to make LLMs more regression-aware. (Lukasik et al., 2024) introduced Regression-Aware Inference for Language models (RAIL), an inference-time technique that applies a Bayes-optimal decision rule to the model's output probabilities. For squared error loss, this rule simplifies to calculating the expected value of the numerical prediction. A follow-up work, Regression-Aware Fine-Tuning (RAFT) by (Lukasik et al., 2025), incorporated this logic directly into the training process. RAFT fine-tunes the model using a squared error loss calculated between the ground truth score and the expected value predictor from RAIL. A follow-up work from (Zausinger et al., 2025) also applied a regression loss over numerical tokens for regression tasks. More recently, TRACT (Chiang et al., 2025) builds on this by combining the numerical scoring ability of RAFT with CoT reasoning. TRACT is a two-stage SFT approach, which 1) first applies RAFT loss on the training dataset, and then 2) uses self-generated CoT from the first-stage trained model, appended with the ground truth labels, to train a base model with the RAFT loss again. The first stage acts as a CoT rewriter, addressing the distribution gap between the training dataset's CoT and the model's own outputs. We note both a flaw in TRACT and a connection between their method and ours: 1. TRACT does not evaluate the quality of the self-generated CoT and treats all generated CoTs as ground truth for the second stage. In contrast, our method evaluates CoT quality using the REAL reward, and 2. In the second stage, TRACT appends the ground truth and then applies RAFT loss. This gradient on the prediction is identical to our prediction update term. The TRACT method can be viewed as a remedy for SFT and an intermediate step between SFT and RL. Our method, REAL, is the first to formalize RL training with regression-aware objectives for the LLM-as-a-Judge setting. Concurrently, PoLi-RL (Song et al., 2026) explores a heuristic listwise reward in the Conditional Semantic Textual Similarity (C-STS) setting (Deshpande et al., 2023; Agirre et al., 2013), which focuses on semantic similarity between texts in specific contexts. While related in spirit, C-STS presents a different and generally more constrained setup compared to LLM-as-a-Judge, which involves broader evaluations based on alignment with task instructions and overall response quality. Additionally, it's important to note that Regression-Aware Fine-Tuning (RAFT) is distinct from Reward-Ranked Fine-Tuning (Dong et al., 2023), which is an alignment method that iteratively fine-tunes models on a filtered subset of high-reward samples. Our work further extends this regression-aware paradigm to reinforcement learning and achieves superior performance in terms of correlation metrics.

**LLM-as-a-Judge**   Recently, employing language models as judges has gained attention as a promising paradigm to mimic the depth and granularity of human evaluation (Gu et al., 2024; Chiang & Lee, 2023; Zheng et al., 2023b; Liu et al., 2023a; Zhu et al., 2024; Ouyang et al., 2022; Zhu et al., 2025). In terms of output format, we classify LLM-as-a-Judge into two types: 1. *Pointwise*, where the output is a score usually ranging from 1 to 5 (Chiang & Lee, 2023; Liu et al., 2023b); 2. *Pairwise*, where the LLM is asked to compare two responses and select the better one (Zheng et al., 2023b; Wang et al.,

2023; Li et al., 2023). While recent advancements such as Prometheus 1 and 2 (Kim et al., 2023; 2024) leverage proprietary models (e.g., GPT-4 (Achiam et al., 2023)) to synthesize training data for standard SFT, they primarily focus on replicating existing evaluator behaviors. Similarly, J1 (Whitehouse et al., 2025) advances the LLM-as-a-Judge paradigm by applying standard RL and transforming existing datasets into prompt-response pairs with verifiable rewards. In contrast, our approach, **REAL**, targets pointwise evaluation and formulates it as a regression problem, directly integrating the continuous nature of scoring objectives into the RL algorithm.

## B. Computational Cost

We provide a detailed computation cost comparison in Tab. 11.

*Table 11.* Computation cost comparison across methods.

| Method | Type | Model Size | Hardware | GPU Hours |
|---|---|---|---|---|
| Prometheus2 | SFT | 7B | 8×A100 (40GB) | 800 |
| TRACT | SFT | 7B | 1×RTX A6000 | 100 |
| Standard RL | RL | 7B | 8×A100 (80GB) | 100 |
| REAL | RL | 7B | 8×A100 (80GB) | 100 |
| REAL | RL | 32B | 16×A100 (80GB) | 500 |

For 7B models, we run RL training for 200 steps, while for 32B models, we train for 500 steps. During gradient updates, REAL uses approximately 55% GPU memory for 7B models and about 90% for 32B models.

We obtain the compute specifications for Prometheus2 and TRACT directly from their original papers. We additionally note that TRACT is a two-stage SFT method requiring two SFT passes plus a full generation pass, where the generation cost is excluded from the reported 100 GPU hours. Meanwhile, Prometheus2 is trained on both pairwise and pointwise datasets using 40GB GPUs, contributing to its substantially higher total GPU-hour requirement.

## C. Extension to pairwise difference modeling

In this section, we demonstrate that REAL can be applied pairwise datasets. In particular, we conducted an additional experiment using Qwen3-1.7B and the HuggingFace `prometheus-eval/Preference_Collection`[1] dataset, which provides scores for Responses A and B, both in the range $[1, 5]$. We construct the pairwise annotation by taking a difference between the scores, and shifting the result by $+5$ to map the final scores to $[1, 9]$. We then prompt the LLM to predict this difference.

In Tab. 12, we evaluate the model on the in-domain Preference-Bench (Kim et al., 2024) (using correlation metrics) and the out-of-domain HHH Alignment (Askell et al., 2021)/ MT-Bench Human Judgement (Zheng et al., 2023a) / Auto-J (Li et al., 2024a) benchmarks (using accuracy with threshold 5, since they do not provide ground-truth score differences). REAL consistently outperforms the SFT baseline on out-of-domain tasks while maintaining strong in-domain performance.

*Table 12.* Results on score-difference prediction using the Preference_Collection dataset. REAL achieves better out-of-domain generalization while maintaining competitive in-domain performance.

| Method | Preference-Bench | | | HHH | MT-Bench | Auto-J |
|---|---|---|---|---|---|---|
| | Pearson ↑ | Spearman ↑ | Acc ↑ | Acc ↑ | Acc ↑ | Acc ↑ |
| Base | 9.30 | 17.6 | 12.2 | 50.7% | 64.8% | 64.7% |
| TRACT | **93.9** | 92.8 | 82.3 | 75.6% | 75.6% | 79.2% |
| REAL | 93.4 | **92.8** | **84.1** | **76.9%** | **76.3%** | **79.7%** |

---

[1] https://huggingface.co/datasets/prometheus-eval/Preference-Collection

*Table 13.* Agreement results on ArenaHard 2.0. We report agreement counts (out of 750) with closed-source judges GPT-4.1 and Gemini-2.5. REAL consistently achieves the strongest alignment across different response-model families.

| Response Model | Judge Model | Agree w/ GPT-4.1 ↑ | Agree w/ Gemini-2.5 ↑ |
|---|---|---|---|
| | Base | 5 | 1 |
| o4-mini-2025-04-16 | TRACT | 71 | 55 |
| | REAL | **81** | **67** |
| | Base | 0 | 2 |
| Gemini-2.5 | TRACT | 56 | 49 |
| | REAL | **81** | **71** |
| | Base | 0 | 2 |
| DeepSeek-R1 | TRACT | 85 | 82 |
| | REAL | **98** | **96** |

# D. Additional benchmarks

In Tab. 13, we provide additional results on Mistral-7B series (Base / TRACT / REAL) on ArenaHard 2.0 (Li et al., 2024b), measuring agreement (out of 750) with closed-source judges (GPT-4.1 and Gemini-2.5). Across three response-model families, REAL consistently achieves the highest agreement, providing evidence that the gains are not limited to older benchmarks.

# E. Proofs of the theoretical results

First, to prove Lemma 3.1, we restate and utilize the following two results from statistical theory: 1) We show that this conditional expectation is the optimal estimator for maximizing Pearson correlation (Lemma E.1); 2) We establish that minimizing the regression-aware loss (MSE) recovers the conditional expectation (Lemma E.2).

## E.1. Additional Lemmas

**Lemma E.1** (Optimality of the Posterior Mean for Pearson Correlation). ***Measure space and generative graph.*** *Let* $(\Omega, \mathcal{F}, \mathbb{P})$ *be a probability space on the tuple* $(x, c, y^*)$ *with the following data-generating process:*

$$x \sim P_{\mathcal{D}} \quad \longrightarrow \quad \begin{cases} c \sim \pi_\theta(\cdot \mid x), \\ y^* \sim P(\cdot \mid x), \end{cases}$$

*i.e.,* $c \perp y^* \mid x$.

*This induces the joint distribution*

$$P_\theta(x, c, y^*) \; = \; P_{\mathcal{D}}(x) \, \pi_\theta(c \mid x) \, P(y^* \mid x), \tag{9}$$

*and expectations* $\mathbb{E}[\cdot]$ *are taken w.r.t.* $P_\theta(x, c, y^*)$ *unless stated otherwise.*

***Conditional mean.*** *Define the conditional mean (posterior mean) of the label given* $(x, c)$ *as*

$$\mu(x, c) \; \triangleq \; \mathbb{E}[y^* \mid x, c]. \tag{10}$$

*Under the conditional independence* $c \perp y^* \mid x$, *we have*

$$P(y^* \mid x, c) = P(y^* \mid x) \quad \text{and thus} \quad \mu(x, c) = \mathbb{E}[y^* \mid x, c] = \mathbb{E}[y^* \mid x]. \tag{11}$$

***Pearson correlation objective.*** *Let* $\hat{y}(x, c) \in \mathbb{R}$ *be any measurable estimator with finite second moment. Assume Pearson correlation is well-defined and non-degenerate:*

$$\mathrm{Var}(y^*) > 0, \qquad \mathrm{Var}(\hat{y}(x, c)) > 0, \qquad \mathrm{Var}(\mu(x, c)) > 0. \tag{12}$$

*Define Pearson correlation as*

$$\rho(U, V) \triangleq \frac{\mathrm{Cov}(U, V)}{\sqrt{\mathrm{Var}(U)\,\mathrm{Var}(V)}}. \tag{13}$$

**Claim.** *The set of estimators that maximize the Pearson correlation $\rho(\hat{y}(x, c), y^*)$ (over all such $\hat{y}$) is exactly the set of positive affine transformations of $\mu(x, c)$:*

$$\hat{y}^*(x, c) \in \arg\max_{\hat{y}} \rho\big(\hat{y}(x, c), y^*\big) \quad \Longleftrightarrow \quad \hat{y}^*(x, c) = a\,\mu(x, c) + b, \ \ \forall a > 0,\ b \in \mathbb{R}. \tag{14}$$

**Remark (invariance and conditional-independence).** Pearson correlation is invariant to positive scaling and shifting: $\rho(\hat{y}, y^*) = \rho(a\hat{y} + b, y^*)$ for any $a > 0$ and $b \in \mathbb{R}$, hence the maximizer is not unique. Moreover, since $c \perp y^* \mid x$ in (9), $\mu(x, c)$ reduces to $\mathbb{E}[y^* \mid x]$ as in (11); we retain the $(x, c)$ notation to match the policy-generated reasoning variable used in post-training. This result follows from Cauchy–Schwarz arguments; see, e.g., (Bottai et al., 2022).

**Lemma E.2** (Optimality of Regression Objective). *The unique global minimizer of the squared error loss $\mathcal{R}(\theta) = \mathbb{E}[(\hat{y}_\theta(x, c) - y^*)^2]$ is the conditional expectation:*

$$\hat{y}_\theta(x, c) = \mathbb{E}[y^* \mid x, c]. \tag{15}$$

This is a standard result in Bayesian decision theory (see (Bishop & Nasrabadi, 2006)).

## E.2. Proof of Lemma 3.1

By Lemma E.2, the estimator that minimizes the squared error loss is $\hat{y}^* = \mathbb{E}[y^*|x, c]$. By setting $a = 1$ and $b = 0$ in Lemma E.1, it follows that this specific estimator is also a maximizer of the Pearson correlation. Thus, the regression objective is a sufficient condition for achieving the optimal correlation estimator. $\square$

## E.3. Proof of Lemma E.1

**Setup and computation graph.** Let $(X, C, Y^*)$ be random variables on a probability space $(\Omega, \mathcal{F}, \mathbb{P})$ with the following data-generating process:

$$X \sim P_\mathcal{D}, \qquad C \sim \pi_\theta(\cdot \mid X), \qquad Y^* \sim P(\cdot \mid X).$$

Equivalently, the conditional independence relation is

$$C \perp Y^* \mid X, \tag{16}$$

which corresponds to the computation graph $C \leftarrow X \rightarrow Y^*$. This induces the joint distribution

$$P_\theta(x, c, y^*) = P_\mathcal{D}(x)\,\pi_\theta(c \mid x)\,P(y^* \mid x). \tag{17}$$

All expectations $\mathbb{E}[\cdot]$ below are with respect to $P_\theta(x, c, y^*)$ unless stated otherwise.

**Conditional mean.** Define the conditional mean (posterior mean)

$$\mu(x, c) \triangleq \mathbb{E}[Y^* \mid X = x, C = c]. \tag{18}$$

Under (16), we have $P(y^* \mid x, c) = P(y^* \mid x)$ and hence

$$\mu(x, c) = \mathbb{E}[Y^* \mid X = x, C = c] = \mathbb{E}[Y^* \mid X = x]. \tag{19}$$

We keep the notation $\mu(x, c)$ because our estimator is explicitly a function of both $(x, c)$.

**Remark (role of the reasoning variable $c$).** In the supervised dataset, only $(x, y^*)$ is observed; the reasoning chain $c$ is *not* a supervised variable and is instead generated by the model/policy via $\pi_\theta(c \mid x)$ during training and inference. From the statistical model (16), conditioning on $c$ does not reveal additional information about the label beyond $x$, so the Bayes-optimal conditional mean reduces as in (19). Nevertheless, in our post-training procedure the predictor we optimize is of the form $\hat{y}_\theta(x, c)$, i.e., it is evaluated on realizations of $(x, c)$ sampled from $P_\mathcal{D}(x)\pi_\theta(c \mid x)$. The lemma therefore characterizes the correlation-optimal form *within the space of functions of* $(x, c)$ (and the correlation is taken over the induced joint distribution over $(x, c, y^*)$).

**Goal.** Let $\hat{y}(x,c)$ be any measurable real-valued estimator with finite second moment. Assume Pearson correlation is well-defined and non-degenerate:

$$\text{Var}(Y^*) > 0, \quad \text{Var}(\hat{y}(X,C)) > 0, \quad \text{Var}(\mu(X,C)) > 0. \tag{20}$$

We show that any maximizer of $\rho(\hat{y}(X,C), Y^*)$ must be of the form $\hat{y}^*(x,c) = a\,\mu(x,c) + b$ with $a > 0$.

**Step 1: Covariance with $Y^*$ reduces to covariance with $\mu(X,C)$.** Recall the Pearson correlation definition

$$\rho(U,V) \triangleq \frac{\text{Cov}(U,V)}{\sqrt{\text{Var}(U)\text{Var}(V)}}. \tag{21}$$

We start from the covariance term:

$$\text{Cov}(\hat{y}(X,C), Y^*) = \mathbb{E}[\hat{y}(X,C)Y^*] - \mathbb{E}[\hat{y}(X,C)]\,\mathbb{E}[Y^*]. \tag{22}$$

Using the law of iterated expectations and conditioning on $(X,C)$,

$$\mathbb{E}[\hat{y}(X,C)Y^*] = \mathbb{E}\big[\mathbb{E}[\hat{y}(X,C)Y^* \mid X,C]\big]. \tag{23}$$

At this point, it is important to note that $\hat{y}(X,C)$ is a (measurable) function of $(X,C)$, hence it is $(X,C)$-measurable and acts as a constant inside the conditional expectation given $(X,C)$. Therefore,

$$\mathbb{E}[\hat{y}(X,C)Y^* \mid X,C] = \hat{y}(X,C)\,\mathbb{E}[Y^* \mid X,C] = \hat{y}(X,C)\,\mu(X,C), \tag{24}$$

and substituting back into (23) yields

$$\mathbb{E}[\hat{y}(X,C)Y^*] = \mathbb{E}[\hat{y}(X,C)\mu(X,C)]. \tag{25}$$

Similarly,

$$\mathbb{E}[Y^*] = \mathbb{E}\big[\mathbb{E}[Y^* \mid X,C]\big] = \mathbb{E}[\mu(X,C)]. \tag{26}$$

Plugging (25)–(26) into (22) gives

$$\begin{aligned}
\text{Cov}(\hat{y}(X,C), Y^*) &= \mathbb{E}[\hat{y}(X,C)\mu(X,C)] - \mathbb{E}[\hat{y}(X,C)]\,\mathbb{E}[\mu(X,C)] \\
&= \text{Cov}(\hat{y}(X,C), \mu(X,C)).
\end{aligned} \tag{27}$$

This identity holds irrespective of whether $Y^*$ depends on $C$; in our setting, $\mu(X,C)$ further reduces to $\mathbb{E}[Y^* \mid X]$ by (19).

**Step 2: Maximizing correlation with $Y^*$ is equivalent to maximizing correlation with $\mu(X,C)$.** Using (27),

$$\begin{aligned}
\rho(\hat{y}(X,C), Y^*) &= \frac{\text{Cov}(\hat{y}(X,C), \mu(X,C))}{\sigma_{\hat{y}}\sigma_{Y^*}} \\
&= \frac{\text{Cov}(\hat{y}(X,C), \mu(X,C))}{\sigma_{\hat{y}}\sigma_{\mu}} \cdot \frac{\sigma_{\mu}}{\sigma_{Y^*}} \\
&= \rho(\hat{y}(X,C), \mu(X,C)) \cdot \frac{\sigma_{\mu}}{\sigma_{Y^*}},
\end{aligned} \tag{28}$$

where $\sigma_{\hat{y}} = \sqrt{\text{Var}(\hat{y}(X,C))}$ and $\sigma_{\mu} = \sqrt{\text{Var}(\mu(X,C))}$. By (20), the factor $\sigma_{\mu}/\sigma_{Y^*}$ is a positive constant that does not depend on the choice of $\hat{y}$. Hence, maximizing $\rho(\hat{y}(X,C), Y^*)$ is equivalent to maximizing $\rho(\hat{y}(X,C), \mu(X,C))$.

**Step 3: Correlation is maximized by a positive affine transformation of $\mu(X,C)$.** Let $\tilde{A} \triangleq A - \mathbb{E}[A]$ denote the centered version of a random variable. Then

$$\text{Cov}(\hat{y}(X,C), \mu(X,C)) = \mathbb{E}\big[\tilde{\hat{y}}\,\tilde{\mu}\big]. \tag{29}$$

By the Cauchy–Schwarz inequality in $L_2$,

$$\mathbb{E}\big[\tilde{\hat{y}}\,\tilde{\mu}\big] \leq \sqrt{\mathbb{E}[\tilde{\hat{y}}^2]}\,\sqrt{\mathbb{E}[\tilde{\mu}^2]} = \sigma_{\hat{y}}\,\sigma_\mu. \tag{30}$$

Dividing both sides by $\sigma_{\hat{y}}\sigma_\mu$ shows

$$\rho(\hat{y}(X,C),\mu(X,C)) \leq 1, \tag{31}$$

with equality if and only if $\tilde{\hat{y}} = a\,\tilde{\mu}$ almost surely for some $a \geq 0$. Equivalently,

$$\hat{y}(X,C) = a\,\mu(X,C) + b \quad \text{a.s.} \tag{32}$$

for some $a \geq 0$ and $b \in \mathbb{R}$. If $a = 0$ then $\text{Var}(\hat{y}(X,C)) = 0$, which is excluded by (20); thus $a > 0$. This proves Lemma E.1.

**Connection to regression objectives.** Since the Bayes-optimal minimizer of expected squared loss is $\mu(x,c) = \mathbb{E}[Y^* \mid x, c]$ (and reduces to $\mathbb{E}[Y^* \mid x]$ under (16)), a regression-aware objective that targets the conditional mean yields an estimator that is also optimal for Pearson correlation, up to a positive affine transformation. A closely related formulation appears in Bottai et al. (2022).

### E.4. Proof of Lemma E.2

**Optimality of MSE for Conditional Expectation.** Let the risk function for the regression-aware loss be defined over the joint distribution of inputs $x$, generated chains $c \sim \pi_\theta(\cdot \mid x)$, and ground-truth labels $y^* \sim P(\cdot \mid x)$:

$$\mathcal{R}(\hat{y}) = \mathbb{E}_{x,c,y^*}\left[(\hat{y}(x,c) - y^*)^2\right]. \tag{33}$$

Let $\mu(x,c) = \mathbb{E}[y^* \mid x, c]$ denote the conditional expectation of the label given the input and reasoning chain. We can decompose the squared error term by adding and subtracting $\mu(x,c)$:

$$\begin{aligned}
\mathbb{E}\left[(\hat{y} - y^*)^2 \mid x, c\right] &= \mathbb{E}\left[(\hat{y} - \mu(x,c) + \mu(x,c) - y^*)^2 \mid x, c\right] \\
&= \mathbb{E}\left[(\hat{y} - \mu(x,c))^2 \mid x, c\right] + \mathbb{E}\left[(\mu(x,c) - y^*)^2 \mid x, c\right] \\
&\quad + 2(\hat{y} - \mu(x,c))\underbrace{\mathbb{E}\left[\mu(x,c) - y^* \mid x, c\right]}_{=0}.
\end{aligned} \tag{34}$$

The cross-term vanishes because $\mathbb{E}[y^* \mid x, c] = \mu(x,c)$ by definition. The total risk is thus:

$$\mathcal{R}(\hat{y}) = \mathbb{E}_{x,c}\left[(\hat{y}(x,c) - \mu(x,c))^2\right] + \text{Var}(y^* \mid x, c). \tag{35}$$

The second term, $\text{Var}(y^* \mid x, c)$, represents irreducible noise and is independent of the estimator $\hat{y}$. Therefore, minimizing the risk $\mathcal{R}(\hat{y})$ is equivalent to minimizing the first term, which is strictly non-negative and equals zero if and only if:

$$\hat{y}(x,c) = \mu(x,c) = \mathbb{E}[y^* \mid x, c]. \tag{36}$$

Thus, the global minimizer of the regression-aware loss is the conditional expectation.

### E.5. Proof of Lemma 4.1

To derive the gradient, we expand the inner expectation as an integral over the reasoning trajectories $c \sim \pi_\theta(c \mid x)$:

$$\mathcal{L}(\theta) = \mathbb{E}_{(x,y^*)\sim\mathcal{D}}\int_c \pi_\theta(c \mid x)r(\theta,x,c)\,dc.$$

Applying the gradient operator $\nabla_\theta$ and utilizing the product rule for differentiation:

$$\begin{aligned}
\nabla_\theta\mathcal{L}(\theta) &= \mathbb{E}_{(x,y^*)\sim\mathcal{D}}\int_c \nabla_\theta\left[\pi_\theta(c \mid x)r(\theta,x,c)\right]dc \\
&= \mathbb{E}_{(x,y^*)\sim\mathcal{D}}\int_c \left[(\nabla_\theta\pi_\theta(c \mid x))\,r(\theta,x,c) + \pi_\theta(c \mid x)\,(\nabla_\theta r(\theta,x,c))\right]dc.
\end{aligned}$$

**Term 1: CoT Policy Update.** Using the log-derivative trick, $\nabla_\theta \pi_\theta(c \mid x) = \pi_\theta(c \mid x) \nabla_\theta \log \pi_\theta(c \mid x)$, the first part of the sum becomes:

$$\mathbb{E}_{(x,y^*)\sim\mathcal{D}} \int_c \pi_\theta(c \mid x) \left[r(\theta,x,c)\nabla_\theta \log \pi_\theta(c \mid x)\right] dc = \mathbb{E}_{(x,y^*)\sim\mathcal{D},c\sim\pi_\theta} \left[r(\theta,x,c)\nabla_\theta \log \pi_\theta(c \mid x)\right].$$

This term performs reinforcement learning on the reasoning process, shifting the policy's mass toward trajectories that yield higher rewards.

**Term 2: Prediction Update.** The second part accounts for the explicit dependence of the reward on $\theta$. Within this integral, the trajectory $c$ is a sample from the current policy (effectively a *stopped-gradient c*):

$$\mathbb{E}_{(x,y^*)\sim\mathcal{D}} \int_c \pi_\theta(c \mid x) \left[\nabla_\theta r(\theta,x,c)\right] dc = \mathbb{E}_{(x,y^*)\sim\mathcal{D},c\sim\pi_\theta} \left[\nabla_\theta r(\theta,x,c)\right].$$

This term enables direct backpropagation through the differentiable components of the reward, mapping the predicted $\hat{y}_\theta(x,c)$ to the target value $y^*$.

# F. Algorithm and Experiments

### F.1. Response Length and Entropy

We note that the response length increases, and entropy decreases throughout REAL training, as shown in Fig. 4.

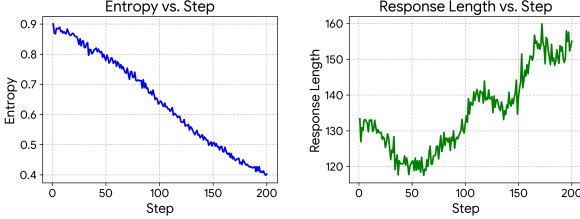

*Figure 4.* Response length and entropy during REAL training. Response length increases, and the per-token entropy of the policy model decreases steadily.

### F.2. Comparison with JEPO

We conduct a direct comparison with JEPO as shown in Tab. 14. The JEPO does not perform as effectively as our proposed approach in regression-aware tasks.

*Table 14.* **Comparison with JEPO with Mistral2-7B.**

| Method | FB Bench | | | FLASK | | | Vic. Bench | | | MT Bench | | | Average | | |
|---|---|---|---|---|---|---|---|---|---|---|---|---|---|---|---|
| | $r$ | $\rho$ | $\tau$ | $r$ | $\rho$ | $\tau$ | $r$ | $\rho$ | $\tau$ | $r$ | $\rho$ | $\tau$ | $r$ | $\rho$ | $\tau$ |
| JEPO | 92.5 | 92.9 | 81.6 | 54.3 | 52.6 | 39.6 | 61.7 | 59.7 | 45.9 | 55.7 | 54.9 | 39.8 | 66.3 | 64.8 | 51.8 |
| REAL | **93.2** | **93.4** | **82.5** | **56.0** | **54.1** | **41.1** | **63.3** | **60.2** | **46.3** | **59.3** | **56.9** | **42.2** | **67.9** | **66.2** | **53.0** |

### F.3. REAL Algorithm

See Algo. 2 for detailed implementation.

# G. Qualitative Results

See below for an example test prompt and REAL's output.

---

**Algorithm 2** REAL: Regression-Aware Reinforcement Learning

---

**Require:** dataset $\mathcal{D}$, policy $\pi_\theta$, log-likelihood weight $\lambda$, prediction weight $\beta$, group size $K$, learning rate $\eta$.
 1: **while** not converged **do**
 2:     Sample batch $B = \{(x, y^*)\} \sim \mathcal{D}$.
 3:     **/* Step 1: Sample $K$ generations with CoTs */**
 4:     Sample $K$ reasoning chains $\{c_1, \ldots, c_K\} \sim \pi_\theta(\cdot|x)$.
 5:     Compute rewards $r_{\text{REAL}}(\theta, x, c_i)$ via Eq. 5.
 6:     **/* Step 2: Compute CoT Update term and Prediction Update term */**
 7:     Compute Advantage $\tilde{A}_{\text{REAL}}^{(i)}$
 8:     Forward the generation again to acquire the gradient for both CoT and Prediction tokens.
 9:     Aggregate terms to estimate $\nabla_\theta \mathcal{L}_{\text{REAL}}^{\text{RLOO}}$ ( Eq. 8).
10:     **/* Step 3: Optimization */**
11:     Update parameters: $\theta \leftarrow \theta + \eta \cdot \frac{1}{|B|} \sum \nabla_\theta \mathcal{L}_{\text{REAL}}^{\text{RLOO}}$.
12: **end while**

---

---

**Sample Question (Vicuna Bench)**

Task Description:

An instruction (might include an Input inside it), a response to evaluate, a reference answer that gets a score of 5, and a score rubric representing a evaluation criteria are given.

1. Write a detailed feedback that assess the quality of the response strictly based on the given score rubric, not evaluating in general.

2. After writing a feedback, write a score that is an integer between 1 and 5. You should refer to the score rubric.

3. The output format should look as follows: "Feedback: (write a feedback for criteria) [RESULT] (an integer number between 1 and 5)

4. Please do not generate any other opening, closing, and explanations.

The instruction to evaluate:

How can I improve my time management skills?

Response to evaluate:

Improving your time management skills can help you to be more productive, reduce stress, and achieve your goals. Here are some tips to help you improve your time management skills:

1. Prioritize your tasks: Make a to-do list at the beginning of each day and prioritize the most important tasks. This will help you to focus on what's most important and avoid wasting time on low-priority tasks.

2. Use a calendar or planner: Use a calendar or planner to schedule your tasks and appointments. This will help you to stay organized and ensure that you have enough time for everything you need to do.

3. Set goals: Set realistic goals for yourself and break them down into smaller, manageable tasks. This will help you to stay focused and motivated.

4. Eliminate distractions: Identify the things that distract you the most, such as social media or email notifications, and find ways to eliminate them. This could be as simple as turning off your phone or logging out of social media during work hours.

5. Take breaks: Taking regular breaks can help you to stay focused and avoid burnout. Take a few minutes every hour to stretch, go for a walk, or do something that you enjoy.

6. Learn to say no: Don't overcommit yourself by taking on too many tasks or responsibilities. Learn to say no to requests that don't align with your goals or values.

7. Review and adjust: Regularly review your schedule and priorities to ensure that you are making the most of your time. Adjust your plans as needed to accommodate changes in your workload or personal life.

By implementing these tips, you can improve your time management skills and become more productive and efficient.

Reference Answer (Score 5):

To improve your time management skills, start by setting clear goals. Understand what you want to achieve and by when. This will give you a roadmap to follow. Next, prioritize your tasks based on their importance and urgency. Use tools like the Eisenhower Box to help you with this. Break down larger tasks into smaller, manageable parts to prevent feeling overwhelmed.

Establish a routine and stick to it. This could be a daily, weekly, or monthly routine depending on your goals. Use tools like calendars, planners, or apps like Google Calendar to schedule your tasks and set reminders.

Avoid multitasking as it can lead to mistakes and inefficiency. Instead, focus on one task at a time until it's complete.

Finally, take regular breaks to avoid burnout. Techniques like the Pomodoro Technique, where you work for 25 minutes then take a 5-minute break, can be very effective. Remember, managing your time effectively also involves taking care of your mental health.

Score Rubrics:

Does the response provide detailed and actionable steps grounded on real-world scenarios?

Score 1: The response lacks detail and provides no actionable steps or real-world context.

Score 2: The response provides limited detail and few actionable steps, with minimal real-world grounding.

Score 3: The response has some detailed and actionable steps, though not all are grounded in real-world scenarios.

Score 4: The response is detailed with mostly actionable steps, and the majority are grounded in real-world scenarios.

Score 5: The response is comprehensive, offering detailed and actionable steps, all of which are firmly grounded on real-world scenarios.

Feedback:

*Table 15.* Example generation by REAL. GPT4 output serves as the reference ground truth answer in the dataset.

| **GPT4 Output:** The response is comprehensive and offers a series of **detailed and actionable steps for improving time management skills**. Steps such as **prioritizing tasks**, using a calendar or planner, setting goals, eliminating distractions, taking breaks, learning to say no, and reviewing and adjusting schedules can be directly implemented by anyone. **These steps are firmly grounded on real-world scenarios**, making them practical and beneficial for individuals looking to enhance their time management skills. | **REAL Output:** The response provided is comprehensive and offers **detailed and actionable steps to improve time management skills.** It covers various aspects such as setting goals, **prioritizing tasks**, creating a schedule, breaking down tasks, eliminating distractions, delegating, using productivity tools, time-blocking, learning to say no, and taking breaks. The response also emphasizes the importance of self-care and mental health, **which is essential in real-world scenarios.** |
|---|---|
| **Score:** 5 | **Greedy decoding:** 5, **RAIL inference: 4.99** |

