# OpenReview forum: "REAL: Regression-Aware Reinforcement Learning for LLM-as-a-Judge"
_ICML.cc/2026/Conference — ICML 2026 regular_

### Official Review · Reviewer_s1jy · 2026-03-05

**Soundness:** 2
**Presentation:** 3
**Significance:** 2
**Originality:** 2
**Overall Recommendation:** 3
**Confidence:** 2

**Summary:**

This paper addresses the mismatch between the regression nature of LLM-as-a-Judge tasks and the binary reward signals used in standard RL post-training. The authors propose REAL, which replaces the binary accuracy reward with a regression-aware reward based on squared error between the model's expected-value prediction and the ground-truth score, augmented with a log-likelihood term on the gold label. A key technical challenge is that this reward depends on the policy parameters, invalidating the standard policy gradient derivation. The authors resolve this using the generalized policy gradient estimator, which decomposes the gradient into two terms: (1) a CoT policy gradient for exploring reasoning trajectories, and (2) a prediction refinement term that directly optimizes the numerical score via backpropagation. The paper also proves that minimizing regression loss is optimal for correlation metrics (Pearson, Spearman). Experiments on Mistral-7B, Qwen3-8B, and Qwen3-32B across four LLM-as-a-Judge benchmarks show that REAL outperforms both regression-aware SFT baselines (RAFT, TRACT) and standard RL (RLOO with accuracy reward), with gains on out-of-domain benchmarks.

**Compliance With Llm Reviewing Policy:**

Affirmed.

**Final Justification:**

The additional ArenaHard 2.0 experiments and computational cost clarification addressed those concerns. However, I still find the core contribution to be a relatively straightforward combination of existing ideas, despite the clean derivation and solid empirical results. I maintain my score.

**Key Questions For Authors:**

1. What is the specific intuition for including λ log π_θ(y*|x,c) in the reward (Eq. 7)? The regression loss alone already optimizes the expected-value prediction toward y*. Does the log-likelihood term serve as a form of regularization, a way to prevent mode collapse in the final-token distribution or something else?
2. The policy-dependent reward requires computing ŷ_θ(x,c) and log π_θ(y*|x,c) during training. While the generation (rollout) phase likely dominates wall-clock time, could you quantify the additional compute cost of REAL relative to standard RLOO?

**Limitations:**

Yes.

**Strengths And Weaknesses:**

### Strengths
1. The paper identifies a genuine mismatch that standard RL treats all incorrect predictions equally, but in pointwise evaluation tasks, the ordinal distance from the ground truth matters.
2. The paper provides a commendably comprehensive set of ablations: group size K, log-likelihood weight λ, dynamic sampling strategy, etc. These help practitioners understand the sensitivity of the method to various design choices.

### Weaknesses
1. Limited novelty of the core technique. The paper's main technical contribution is applying the generalized policy gradient to handle policy-dependent rewards. The reward itself (Eq. 7) is essentially the negative squared error plus a log-likelihood term on the gold answer. The log-likelihood component is structurally similar to a KL penalty toward ground-truth tokens, and the overall formulation feels like a relatively straightforward combination of the RAFT/TRACT regression loss with a standard policy gradient.
2. The reward design intuition is unclear. The reward in Equation 7 includes λ log π_θ(y*|x,c), which is the log-probability of the ground-truth final answer under the current policy. The intuition for including this term alongside the regression loss is not well-explained. On one hand, the regression loss already encourages the expected-value prediction to match y*; on the other hand, the log-likelihood term directly pushes the model to assign high probability to the exact token y*. A clearer discussion of why this combination works and when it might conflict would strengthen the paper.
3. The evaluation benchmarks (Feedback Bench, FLASK, Vicuna Bench, MT Bench) are from 2023 and primarily involve older model outputs (GPT-3.5, Vicuna-13B, Llama-2-Chat-13B). The LLM-as-a-Judge landscape has evolved substantially since then, with newer and more challenging benchmarks (e.g., ArenaHard 2.0, WildBench, JudgeBench).
4. Narrow scope of "regression" setting. The paper frames its contribution broadly as "regression-aware RL," but the actual experimental scope is limited to pointwise scoring on a 1–5 scale with integer ground truths. This is a very constrained regression problem. The paper does not evaluate on tasks with continuous scores, wider score ranges.

---

> ### Author Rebuttal · Authors · 2026-03-30
>
> > w1. On novelty
>
> We clarify the substantial novelty of our work, which can be viewed at three distinct levels:
>
> 1.**Principled Evolution from SFT to RL**: Overall, REAL extends prior regression-aware SFT methods into the active RL regime to enable CoT exploration, while naturally introducing nontrivial theoretical and algorithmic consequences, including correlation-optimality and effective training under policy-dependent rewards (explained in detail in the two points below). This is echoed by **Reviewer aoDT**, who notes the paper *"considers a very important problem,"* and **Reviewer o7Cv**, who remarks that *"shifting the training of 'regression-aware' LLMs from static SFT to active RL represents a highly logical and impactful next step for this field."*
>
> 2.**Methodological Novelty:** As the reward depends explicitly on $\pi_\theta(k\mid x,c)$, the reward becomes **policy-dependent**, meaning standard policy-gradient derivations do not directly apply. We derive a **generalized-policy-gradient** (Lemma 4.1) formulation that decomposes into two distinct, non-trivial terms: a **CoT exploration** term and a **Prediction refinement** term. To our knowledge, this decomposition has not previously been formulated for judge-model training. This is echoed by **Reviewer aoDT**, who notes the generalized policy gradient *"is a clever way"*, and **Reviewer o7Cv**, who remarks that this is *"an exceptionally elegant decomposition"*.
>
> 3.**Theoretical Novelty:** We address the fundamental mismatch between training and evaluation. Since judge models are typically evaluated via list-wise correlation metrics, we provide a theoretical analysis showing that our **regression objective shares the same optimal estimator as correlation** (Lemma 3.1 & 3.2), providing a principled route for RL optimization in this setting. This is echoed by **Reviewer o7Cv**, who remarks that *"the mathematical formalization of this problem is excellent. "*
>
> > W2&Q1. On the log-likelihood term
>
> We kindly refer the reviewer to our response to **Reviewer o7Cv (Q2. On log-likelihood)**.
>
> > W3. More benchmarks
>
> Our choice of Feedback Bench, FLASK, Vicuna Bench, and MT Bench was mainly motivated by **controlled and fair comparison** with the most relevant prior methods, including Prometheus/TRACT. Using the same benchmark suite allows us to **isolate the effect of the proposed algorithm rather than confounding it with changes in datasets or evaluation protocols**. Besides, newer benchmarks such as JudgeBench emphasize reasoning-heavy domains such as math and coding, which **introduce a different evaluation regime from our training dataset and can make it harder to disentangle algorithmic improvements from domain shift.**
>
> To further address the reviewer’s concern, we additionally evaluated the **Mistral-7B series** (Base/TRACT/REAL) on **ArenaHard 2.0**, measuring agreement (out of 750) with closed-source judges (GPT-4.1 and Gemini-2.5). Across three response-model families, REAL consistently achieves the highest agreement, providing evidence that the gains are not limited to older benchmarks.
>
> | Response model | Judge model | Agree with GPT-4.1 | Agree with Gemini-2.5 |
> |-|-|-|-|
> | o4-mini-2025-04-16 | Base | 5 | 1 |
> | o4-mini-2025-04-16 | TRACT | 71 | 55 |
> | o4-mini-2025-04-16 | REAL | **81** | **67** |
> | Gemini-2.5 | Base | 0 | 2 |
> | Gemini-2.5 | TRACT | 56 | 49 |
> | Gemini-2.5 | REAL | **81** | **71** |
> | DeepSeek-R1 | Base | 0 | 2 |
> | DeepSeek-R1 | TRACT | 85 | 82 |
> | DeepSeek-R1 | REAL | **98** | **96** |
>
> > W4. The scope of our setting
>
> We kindly refer the reviewer to our response to **Reviewer o7Cv (W2. Generalizability to other settings)**.
>
> > Q2. On computational cost
>
> We believe there is a misunderstanding regarding the compute overhead. REAL has essentially the **same computational cost as standard RLOO**.
>
> In the VERL framework, after rollout produces a sampled trajectory $[c,y]$, the model is typically run once more on the full sequence $[x,c,y]$ in order to compute gradients of log likelihood of the generated tokens. This step is also required by standard RL methods such as RLOO. Importantly, this same forward pass also returns the token logits at each position on the whole vocab. Therefore, from the existing logits, we can directly obtain both the probability of the ground-truth score token, $\pi_\theta(y^*|x,c)$ (as the ground-truth label $y^\*$ is already known, $\pi_\theta(y^\*|x,c)$ can be extracted directly from the corresponding vocabulary index), and the full score-token distribution $\{\pi_\theta(k \mid x,c)\}$.
>
> We also kindly refer the reviewer to our response to **Reviewer aoDT (Q3: Computational Cost)**.
>
> **We believe that our point-to-point response with newly-added experiments and clarification could greatly address the reviewer’s concerns and help situate our algorithmic work within the broader progress on regression-aware RL regimes. If so, we would be grateful if the reviewer would consider increasing their score.**

---

> > ### Author Rebuttal · Reviewer_s1jy · 2026-04-03
> >
> > I acknowledge the authors' rebuttal. The additional ArenaHard experiments address my concern about benchmark recency, and the clarification on computational cost is satisfactory.
> >
> > While I still have reservations about the degree of novelty, since the core technique remains a relatively natural combination of existing ideas. Given my limited familiarity with this specific line of work, I will keep my score.

---

> > > ### Author Response · Authors · 2026-04-03
> > >
> > > Dear Reviewer s1jy,
> > >
> > > We sincerely thank you for your time spent reviewing on our work.
> > >
> > > As for your remaining concern, considering the limited familiarity mentioned by the reviewer, we would like to briefly clarify why our work is valuable to the community and provide more context for our work:
> > >
> > > 1.**"Apparent Naturalness" as a Strength:** We consider this **apparent naturalness** and principled transition from SFT to RL to be a hallmark of our solution, as **it is clean, intuitive, yet highly effective**. As shown in Table 1 of our paper, our method (on Mistral2-7B, Qwen3-8B, Qwen3-32B) empirically provides strong improvements over prior regression-aware fine-tuning works in LLMs: RAIL [1], RAFT [2], and TRACT [3].
> > >
> > > 2.**Algorithmic Novelty vs. Concurrent Work:** To our knowledge, the most recent work targeting RL in the LLM-as-a-Judge setting is **J1** [4]. They expand the LLM-as-a-Judge RL training dataset by converting all judgment tasks into verifiable formats (i.e., binary rewards in our context), and then **simply apply standard off-the-shelf GRPO**. In contrast, our work directly targets the regression nature of LLM-as-a-Judge (e.g., when the ground truth is 5, a prediction of 4 is significantly better than 1). In addition, **we are the first to show that standard RL fails to improve over advanced regression-aware SFT methods, as presented in Fig. 2.** We hope our findings can help the community rethink the regression nature of the LLM-as-a-Judge task, which differs from the binary setting in math RL methods.   **We believe this brings algorithmic novelty and sheds light on the regression nature of training in LLM-as-a-Judge**.
> > >
> > > 3.**Theoretically Grounded Derivation:** We hope you find our rigorous derivation from the regression-aware objective—with the help of the generalized policy gradient, which decomposes into the CoT update and Prediction Refinement—interesting.
> > >
> > > Therefore, we believe our work is valuable to both **(1) regression-aware fine-tuning in LLM** and **(2) large-scale LLM-as-a-Judge training algorithms.**
> > >
> > > We hope this clarifies our position and increases your confidence in our work. Thank you again for your time. Feel free to let us know what you think.
> > >
> > > ---
> > >
> > > [1] RAIL: Regression Aware Inference with LLMs. EMNLP 2024.
> > >
> > > [2] RAFT: Better Autoregressive Regression via Regression-aware Fine-tuning. ICLR 2025.
> > >
> > > [3] TRACT: Regression-Aware Fine-tuning Meets Chain-of-Thought Reasoning for LLM-as-a-Judge. ACL 2025.
> > >
> > > [4] J1: Incentivizing thinking in llm-as-a-judge via reinforcement learning. ICLR 2026.

---

### Official Review · Reviewer_o7Cv · 2026-03-12

**Soundness:** 3
**Presentation:** 3
**Significance:** 3
**Originality:** 3
**Overall Recommendation:** 4
**Confidence:** 4

**Summary:**

The paper proposes REAL (Regression-Aware Reinforcement Learning), a novel post-training framework designed specifically for "LLM-as-a-Judge" applications. Overall, this study discusses a broad theme of effectively aligning Large Language Models for evaluation tasks where outputs possess an inherent ordinal or continuous structure (e.g., scoring from 1 to 5). While standard RL algorithms rely on binary rewards (like accuracy), which ignore the magnitude of errors, and previous regression-aware methods (like RAFT and TRACT) are limited to the Supervised Fine-Tuning (SFT) paradigm, REAL bridges this gap. The authors formulate a policy-dependent regression reward and utilize a generalized policy gradient to decompose the optimization into two parts: Chain-of-Thought (CoT) exploration and final prediction refinement. Empirical results across models up to 32B parameters demonstrate state-of-the-art performance, particularly in out-of-domain correlation metrics.

**Compliance With Llm Reviewing Policy:**

Affirmed.

**Final Justification:**

All my concerns have been addressed, so I will maintain my positive review.

**Key Questions For Authors:**

# Questions

### 1. Training Cost:

Could you provide a brief analysis of the computational overhead of REAL training (e.g., GPU hours or relative time factor) and compare it with TRACT and standard RL?

### 2. Log-Likelihood Weight (λ):

In Table 4, λ = 0 (pure regression reward) significantly underperforms λ = 1. If the theoretical proofs rely on MSE/regression risk, what is the exact mechanism by which the standard next-token prediction loss on the final token regularizes or improves the correlation? Would the model forget how to correctly format numbers without it?

### 3. RL Setup:

In Algorithm 2, is a KL-divergence penalty against a reference model applied during the RL update (as is standard in RLHF/RLAIF), or is the optimization driven solely by the REAL objective and RLOO clipping? If no KL penalty is used, did you observe any reward hacking in CoT generation during longer training steps?

**Limitations:**

# Limitations

### 1. Limited to Pointwise Regression Settings:

The current framework is primarily formulated for pointwise regression tasks (i.e., predicting a specific continuous score, such as 1–5) and does not directly address or apply to pairwise preference learning scenarios (i.e., comparing two responses and selecting the better one).

### 2. Reliance on Self-Generated CoT:

This method depends on the base model’s own generated reasoning process (Chain-of-Thought), which may cause the model to inherit or amplify the base model’s inherent biases or systematic errors.

**Strengths And Weaknesses:**

# 1. Strong Theoretical Foundation

The mathematical formalization of this problem is excellent. The authors successfully prove that:

- Minimizing the regression loss yields the Bayesian optimal estimator of the Pearson correlation coefficient (Lemmas 3.1 & 3.2).
- Furthermore, the rigorous derivation of the generalized policy gradient for the policy-dependent reward function (Lemma 4.1) leads to an exceptionally elegant decomposition:
  - Inference (CoT update)
  - Scoring (prediction update)

---

# 2. Novelty and Significance

Shifting the training of “regression-aware” LLMs from static SFT to active RL represents a highly logical and impactful next step in this field.

Furthermore, the paper points out:

- Traditional RL treats inference and the final answer as homogeneous structures;
- REAL, however, explicitly distinguishes their structural differences.

This conceptual separation is a meaningful contribution to the field.

---

# 3. Comprehensive Experimental Validation

The experimental design is well-structured and rigorous:

- Tests were conducted across different tokenizer architectures (Mistral and Qwen);
- Scaled up to 32B-parameter models;
- The method’s robustness and scalability were validated.

Furthermore, the performance improvements on out-of-distribution generalization tasks (such as MT Bench and FLASK) are highly convincing.

---

# 4. High Practicality

The ablation experiments are very thorough.

In particular:

- REAL uses single-shot generation (N=1) during inference;
- Achieves peak efficiency within the RAIL framework;
- Avoids the high latency associated with multi-shot averaging.

This offers significant advantages in real-world deployment.

---

# Weaknesses

## 1. Insufficient Discussion of Computational Overhead

Although inference efficiency is high, during the training phase:

- Each prompt still requires sampling K trajectories (e.g., K = 8 or 16).

The paper does not directly compare:

- REAL
- Standard RL
- Two-stage SFT methods (TRACT)

in terms of training costs (time or FLOPs).

A more systematic analysis of training costs would make the paper more comprehensive.

---

## 2. Generalizability to Pairwise Settings

This work primarily focuses on pointwise scoring.

However, modern reward modeling and LLM evaluation typically rely on:

- Pairwise preference comparisons
- The Bradley–Terry model

Although the authors mention this in the limitations section, a brief discussion would be helpful regarding:

- Whether REAL can be extended to predict continuous margins between responses;
- Or whether it is suitable for pairwise difference modeling.

---

## 3. Issues with Baseline Comparisons

The paper uses RLOO for stabilized training.

Although RLOO is effective, in inference-based RL tasks:

- Group Relative Policy Optimization (GRPO)

has become the standard method (the authors also cite it in DeepSeekMath / R1).

Therefore, it is recommended that:

- The authors explain why they chose RLOO over GRPO;
- Or provide a brief comparative experiment.

---

> ### Author Rebuttal · Authors · 2026-03-30
>
> > W1&Q1. Computational Cost
>
> We kindly refer the reviewer to our response to **Reviewer aoDT (Q3: Computational Cost)**.
>
> > W2. Generalizability to other settings
>
> 1. Extending to continuous scores?
>
> - **For Inference**, REAL already supports continuous-valued prediction. REAL produces a continuous RAIL predictor $\hat y_\theta = \sum_{k} k \pi_\theta(k|x,c)$. In fact, **REAL is already evaluated on continuous-valued benchmarks** (e.g., FLASK, MT-Bench, and Vicuna), where the gt values $y^*$ are averages of 3 integer labels from humans or GPT-4.
> - **For training,** most widely-used pointwise judge-model training datasets only provide integer scores, such as those in Prometheus1/2, and JudgeLM. However, **this is a property of the dataset rather than a limitation of our method**. REAL can directly optimize for continuous $y^\*$ using the $(y^\* - \hat y_\theta)^2$ term.
> - **Dealing with wider range:** For datasets with score ranges exceeding our default scale (e.g., $y^\* > 9$), labels can be normalized to $[0, 9]$ during training and mapped back via de-normalization at inference.
>
> 2. Extending to pairwise difference modeling?
>
> When only score differences are available, REAL can be applied directly. We conducted an additional experiment using **Qwen3-1.7B** and the Huggingface `prometheus-eval/Preference_Collection` dataset, which provides scores for Response A and B in the range $[1, 5]$. We calculate the difference and shift it by $+5$ to map the final scores to $[1, 9]$. We then prompt the LLM to predict this difference.  We evaluated the model on the in-domain Preference-Bench (using correlation metrics) and the out-of-domain HHH alignment/MT Bench/Auto-J (using accuracy with a threshold 5, as they don't provide gt score difference). REAL consistently outperforms the SFT baseline on out-of-domain tasks while maintaining high in-domain performance.
>
> ||P-Bench |P-Bench|P-Bench| HHH| MT-Bench|Auto-J|
> |-|-|-|-|-|-|-|
> || $r$| $\rho$ | $\tau$ |acc|acc|acc|
> | Base| 9.30| 17.6| 12.2 | 50.7%| 64.8%  | 64.7%|
> | TRACT|**93.9**|92.8| 82.3| 75.6%| 75.6%| 79.2%|
> | REAL| 93.4 | **92.8**| **84.1**| **76.9%**| **76.3%**|**79.7%**|
>
> > W3. On GRPO
> 1. First, We kindly refer the reviewer to our response to  **Reviewer aoDT (Q2)** for a detailed discussion on why standard RL (like PPO/GRPO/RLOO) fails.
> 2. Second, we add a **GRPO baseline** using Mistral2-7B:
> |Method|Type|$r$|$\rho$|$\tau$|
> |-|-|-|-|-|
> |Base|None|52.1|50.8|39.5|
> | **GRPO (init from Base)**|RL|56.3|54.2|42.7|
> |TRACT|SFT|63.2|62.2|49.8|
> |**GRPO (init from TRACT)**|RL|63.5|62.5| 49.9|
> |REAL|RL|**67.9**|**66.2**|**53.0**|
>
> > Q2. On log-likelihood
>
> 1.  Would the model forget how to correctly format numbers without log-likelihood?
>
> **No.** Using the regression-aware reward alone ($\lambda=0$) already achieves strong SOTA performance, improving Pearson  by 2.8 points. Setting $\lambda=1$ yields a further improvement (+1.9 Pearson), while larger values (e.g., $\lambda=5/10$) mainly lead to a plateau rather than substantial additional gains, as shown in Tab. 4. This suggests that **the regression-aware objective is the primary driver of performance, and the log-likelihood term acts as a beneficial complement rather than the main source of improvement.**
>
> 2. On the mechanism for log-likelihood term
>
> We observe that setting $\lambda > 0$ **accelerates** RL convergence and improves final performance by providing a critical **distributional constraint**: While a standalone MSE loss only ensures that the expected value $\hat y_\theta$ is close to the ground truth $y^\*$, it does not explicitly regularize the shape of the token distribution $\pi_\theta(y|x, c)$. By incorporating the log-likelihood term, we force the model to place reasonable probability mass directly on the ground-truth label. Intuitively, this prevents the optimization from settling into unreasonable distributional shape, thereby speeding the training process.
>
> > Q3. On KL penalty
>
> We do not include a KL divergence penalty in our experiments because:
> 1. In our experiments, we find the KL penalty hindered RL convergence, causing correlation metrics to stagnate. This aligns with findings in the **DAPO paper (Sec. 2.3)**: While KL penalties are standard in RLHF to ensure user experience and avoid reward hacking, we find that in **outcome-oriented tasks** (e.g., math, coding, or our scoring setting), allowing the policy to deviate freely from its initialization enables the model to explore effective reasoning trajectories, which are critical for superior performance.
> 2. Despite the exclusion of the KL term, our results on diverse out-of-domain benchmarks including MT-Bench, Vicuna, and FLASK demonstrate that REAL achieves robust generalization rather than over-fitting to the training dataset.
>
> **We sincerely thank you for your positive comments on our paper, and hope our responses address your concerns and help increase your confidence in our work. We are happy to offer more clarification if needed.**

---

> > ### Author Rebuttal · Reviewer_o7Cv · 2026-04-04
> >
> > Thanks for rebuttal, I will keep my positive score.

---

> > > ### Author Response · Authors · 2026-04-04
> > >
> > > Dear Reviewer,
> > >
> > > Thank you very much for your positive review. We are glad to see that your concerns have been addressed.
> > >
> > > Best，
> > >
> > > Authors

---

### Official Review · Reviewer_aoDT · 2026-03-17

**Soundness:** 3
**Presentation:** 3
**Significance:** 3
**Originality:** 3
**Overall Recommendation:** 5
**Confidence:** 3

**Summary:**

Problem: training LMs to excel in LM-as-a-judge settings is hard. Traditional methods (e.g. Prometheus) use an SFT loss. This traditionally treats every categorical score as distinct (ignoring the ordinal nature of performance ratings), though this has been recently fixed by Regression-Aware Fine-tuning. Moreover, these days there is a desire to use RL for learning to reason during judge training, for learning to reason. When rewarding a judge, binary rewards suffer from the same non-ordinality issue.

Ideally one could reinforce the judge’s ability to attain a high correlation with ground-truth scores. But this is not practical because correlation is a population statistic, and not suitable for a point-level reward.

The paper suggests that a standard squared-error regression loss is optimal for this task. They design an algorithm to execute this, and show excellent correlation with human judgements on a number of benchmark datasets.

**Compliance With Llm Reviewing Policy:**

Affirmed.

**Final Justification:**

This paper tackles an interesting problem (of learning-to-judge via RL). They devise a principles-based learning algorithm for this task, and this method clearly works well in practice. Therefore, I think it is worthy of publication.

**Key Questions For Authors:**

- Above, I wrote that “I think the framing of this as a ‘theoretical challenge’ is confusing, since one could still use traditional policy-independent-reward algorithms like GRPO. I think the use of generalized policy gradients is a clever way to take advantage of the fact that rewards here are an exact function of the policy’s behavior, but this isn’t strictly necessary.” Do you agree with my understanding or am I wrong?
- Is “RLOO (Acc)” just using RLOO with a squared error reward, or does it include other changes or nuances (e.g. the use of the generalized policy gradient rather than using the standard REINFORCE policy gradient of RLOO)
- How does the computational cost of the proposed method (RL w/ squared rewards with generalized policy gradient) compare to prior works like TRACT or Prometheus?

**Limitations:**

The limitations are not bad but they are too brief, and I hope the authors will expand upon these if accepted

**Strengths And Weaknesses:**

**Soundness:**

*Positives:*

- The proposed method clearly works well in practice against very strong baselines
- The ablations are well-designed and provide increased confidence that this method does what it sets out to do

*Negatives*

- I’d like to see a simpler baseline of simply using the squared error reward (”regression-aware reward”). I think this is simpler than the proposed “Generalized Policy Gradient” approach, because you can just use an algorithm like RLOO using the squared error directly as the reward. I believe “RLOO (Acc)” is not exactly this.

**Presentation:**

*Positives:*

- The paper is, for the most part, very well written
- The use of math seems considered and improves the reader’s understanding

*Negatives:*

- The intro motivates the paper’s contributions via “integrating regression-aware objectives into RL introduces a significant theoretical challenge: the resulting reward functions depend explicitly on the policy parameters themselves. To resolve the resulting policy-dependency reward issue, we employ the generalized policy gradient (Schulman et al., 2015) that explicitly accounts for parameter-dependent rewards.”  — I didn’t feel this was clear at all
    - This makes more sense after reading 4.1, but I think the framing of this as a “theoretical challenge” is confusing, since one could still use traditional policy-independent-reward algorithms like REINFORCE or PPO. I think the use of generalized policy gradients is a clever way to take advantage of the fact that rewards here are an exact function of the policy’s behavior, but this isn’t strictly necessary.
- Lemma 3.1 is a bit confusing — how is list-wise Pearson correlation $\rho(\hat{y}(x, c), y^∗ )$ well-defined if $y^*$ is a fixed label

**Significance:**

*Positives:*

- This paper considers a very important problem at the moment
- This paper offers an intuitive and well-designed solution
- The provided ablations help the reader identify which aspects are most crucial (e.g. squared error loss) and which are still useful but not as crucial (e.g. the log-likelihood term)

*Negatives:*

- The paper does not describe computational costs — neither in absolute terms (e.g. training took 8 hours on 8 NVIDIA A100 GPUs). Presently the paper describes the hardware used but not how exactly it was used.

**Originality:**

*Positives:*

- The method puts together some familiar ideas into a cohesive algorithm that shows clear value on a task that people care about. That is a

*Negatives:*

- Motivation is a little unclear. Isn’t RL always “regression-aware” if you use a reward directly?  The paper contrasts the proposed method with RL using a binary accuracy reward, but that is just a specific (and relatively narrow) formulation of RL
    - The paper’s solution of using a squared-error regression loss as a reward is the obvious solution to the presented problem

---

> ### Author Rebuttal · Authors · 2026-03-30
>
> **Notation: We use $\hat y$ to refer to the label generated by standard decoding, and $\hat y_\theta = \sum_{k} k \pi_\theta(k|x,c)$ to denote the RAIL prediction.**
>
> > Clarification on $ y^*$
>
> We clarify that in Lemma 3.1 $y^\*$ is a random variable with a distribution $P(y^*|x)$ rather than a fixed label (see Lemma 3.1 proof in Section B.1). We will improve notation and clarify this in the updated draft.
>
> > Q2: On RLOO(acc) and RLOO with regression-aware reward
>
> The RLOO(acc) baseline in our paper uses standard REINFORCE without a generalized policy gradient. Its reward is a binary 0-1 indicator, i.e., $\mathbf{1}\\{\hat y = y^*\\}$.
>
> We understand the reviewer's suggestion to use a **policy-independent regression-aware reward** combined with standard RL. To address this, we formulated a reward using the squared error, $-(\hat y - y^*)^2$, and tested it as an additional baseline denoted as **RLOO(reg)**. As shown in the detailed results below, its behavior is very similar to RLOO(acc) and it collapses quickly.
>
> | Method| Type | $r$ | $\rho$ | $\tau$ |
> |-|-|-|-|-|
> |TRACT| SFT |63.2 | 62.2 | 49.8 |
> | RLOO (acc) | RL|64.1|62.7|50.3|
> |**RLOO (reg)**|RL|63.4|62.6| 50.1|
> |REAL|RL|**67.9**|**66.2**|**53.0**|
>
> One might get confused at why RLOO(reg) struggles with regression problems. To clarify, we provide a concrete example observed in our experiments.  **Intuitively, the issue lies in standard RL failing to evaluate the true quality of sampled trajectories**.
>
> We note that we initialize our REAL from the SFT ckpt, which already achieves a $>0.9$ Pearson on the in-domain Feedback-Bench. This means that, strictly through SFT, the model correctly predicts most labels.
> For instance, if the ground truth $y^*$ is 5, most rollouts will correctly output a discrete 5. Under standard RL frameworks (whether RLOO, GRPO, or PPO), all trajectories that output 5 are treated as having the exact same quality and yield the same advantage. Consequently, standard RL cannot distinguish between a rollout that predicts 5 with a probability of 0.9 and one that predicts 5 with a probability of 0.8. Even with a policy-**independent** regression-aware reward, the resulting advantage remains identical as long as the discretely predicted label $\hat y$ is the same. This strips the model of an effective learning signal to improve its performance, a failure mode we observed frequently during our RLOO(acc) and RLOO(reg) experiments.
>
> Our REAL method circumvents this by using $\hat y_\theta$, which captures the distribution of the predicted label. This provides a fine-grained and highly effective learning signal. In the previous scenario, $\hat y_\theta$ might evaluate to 4.99 versus 4.70, resulting in distinctly different advantages that properly guide the model.
>
> We believe the illustration above highlights the significance of both our REAL reward formulation and our use of the generalized policy gradient. We sincerely thank the reviewer for prompting this insightful addition.
>
> > Q1: Framing of the technical challenge and the necessity of generalized policy gradient
>
> In short, per the explanation above, we believe the generalized policy gradient is essential because the REAL reward explicitly depends on the policy parameters. The REAL reward can distinguish between trajectories that yield the same standard decoded label $\hat y$, enabling improvements over advanced SFT methods. In contrast, RL with a policy-independent reward fails to achieve a fine-grained distinction among predictions with varying distributions over targets, as evidenced by our RLOO(acc) and RLOO(reg) experiments above.
>
> > Q3. Computational Cost
>
> We provide a detailed computation cost comparison in the table below and thank the reviewer for highlighting this point.
>
> | Method | Type | Model Size | Hardware | GPU Hours |
> | :--- | :--- | :--- | :--- |-|
> | **Prometheus2** | SFT |7B | 8 x NVIDIA A100 (40GB) | 800 |
> | **TRACT** | SFT| 7B | 1 x NVIDIA RTX A6000 | 100 |
> |**RLOO(acc)**| RL | 7B | 8 x NVIDIA A100 (80GB)| 100 |
> | **REAL** | RL | 7B | 8 x NVIDIA A100 (80GB) | 100 |
> | **REAL** | RL |32B | 8 x 2 NVIDIA A100 (80GB) | 500 |
>
> For 7B models, we run RL training for 200 steps, while for 32B models, we train for 500 steps. During gradient updates, REAL uses approximately 55% of GPU memory for 7B models and about 90% for 32B models.
>
> We obtained the compute specifications for Prometheus2 and TRACT directly from their respective original papers. It is important to note the following context for these baselines:
> - TRACT is a two-stage SFT method that requires two SFT passes plus a full generation pass (excluded from the 100h).
> - Prometheus2 was trained on both pairwise and pointwise datasets using 40GB GPUs, which contributes to its significantly higher total GPU hour requirement.
>
> **We sincerely thank the reviewer for the insightful and helpful suggestions, and hope our responses address your concerns and help increase your confidence in our work.**

---

> > ### Author Rebuttal · Reviewer_aoDT · 2026-04-04
> >
> > Thank you for the helpful response.
> >
> > > Q2: On RLOO(acc) and RLOO with regression-aware reward
> >
> > This is an excellent and compelling experiment. Thank you for sharing this.
> >
> > > Q1: Framing of the technical challenge and the necessity of generalized policy gradient
> >
> > Thank you for helping me better understand the novelty gap here. Overall I think this mostly addresses my fixable concerns and I will increase my score to a 5 accordingly.

---

> > > ### Author Response · Authors · 2026-04-04
> > >
> > > Dear Reviewer,
> > >
> > > We thank you so much for your positive and helpful review! We are glad to see your concerns have been addressed.
> > >
> > > Best,
> > >
> > > Authors

---

### Decision · Program_Chairs · 2026-04-30

**Decision:**

Accept (regular)

**Comment:**

REAL presents a principled regression-aware RL framework for LLM-as-a-Judge, targeting the mismatch between ordinal scoring tasks and standard binary-reward RL. Reviewers found the problem important and timely, and highlighted the method’s strong technical foundation, elegant generalized policy-gradient decomposition, and solid empirical performance across scales and benchmarks (aoDT, o7Cv). The experimental study was viewed as rigorous and practically relevant, with strong ablations and convincing out-of-domain gains (aoDT, o7Cv, s1jy).

Most concerns focused on novelty, compute, and scope, but these were substantially addressed in discussion. The added RLOO(reg) experiment was especially persuasive and helped clarify the novelty and necessity of the proposed formulation (aoDT), while additional results on GRPO, ArenaHard 2.0, compute cost, and extensions beyond the core setting resolved or largely addressed the remaining questions (aoDT, o7Cv, s1jy). Although s1jy maintained some reservation about novelty, this was balanced by the paper’s clear empirical strength and the positive post-rebuttal consensus. Overall, I recommend acceptance.